



# Comparison and evaluation of updates to WRF-Chem (v3.9) biogenic emissions using MEGAN

Mauro Morichetti[1], Sasha Madronich[2], Giorgio Passerini[3], Umberto Rizza[1], Enrico Mancinelli[3], Simone Virgili[3] and Mary Barth[2]

[1]Institute of Atmospheric Sciences and Climate, National Research Council of Italy, Unit of Lecce, Italy
[2]National Center for Atmospheric Research, Boulder, Colorado, USA
[3]Department of Industrial Engineering and Mathematical Science, University of Polytechnic of Marche, Ancona, Italy

*Correspondence to* Mauro Morichetti (m.morichetti@isac.cnr.it)

**Abstract.** Natural gases produced by the Earth's ecosystem include a wide range of volatile compounds such as isoprene, monoterpene, nitric oxide, carbon monoxide, and other non-methane volatile organic compounds: the so-called biogenic volatile organic compounds (BVOCs). BVOCs are highly reactive and thus can impact air quality and aerosol radiative forcing. BVOC emission fluxes have been consistently included in global and regional chemical transport models (e.g., Model of Emissions of Gases and Aerosols from Nature, MEGAN). However, present climate models still have large uncertainties in estimating biogenic trace gases. These uncertainties result from several factors, including uncertainties in emission activity factors that are controlled by environmental conditions, specification of vegetation type, and plant emission factors. This work concerns the evaluation and test of a set of updates made to MEGAN, a model for estimating fluxes of biogenic compounds between terrestrial ecosystem and the atmosphere, which is embedded into the Weather Research and Forecasting model coupled with chemistry (WRF-Chem version 3.9). Two different test cases are presented, the first covering Europe, and the second for a domain in the Southeast United States. Our study considers four simulations for each update made to MEGAN, (i) a control run with no changes to MEGAN; (ii) a simulation with the emission activity factors modified following MEGAN version 2.10; (iii) a simulation considering the changes to the plant functional type emission factor; (iv) a simulation with the isoprene emission factor calculated within the MEGAN module (the emission factor of isoprene is obtained from the input database directly in WRF-Chem). For the Europe domain region, a sensitivity study on BVOC emissions was performed for a high-ozone episode in August 2015. The updated MEGAN model led to significant increases, by factors of 2 or more, of the estimated BVOC emissions. The comparison of WRF-Chem results for the European domain with experimental data from the Airbase web-portal (European air quality database) showed that the temporal and spatial distribution of ozone are well represented. However, comparing the updated MEGAN simulations with the control run, ozone concentration bias increased substantially. Results from the U.S. domain are compared with the Nitrogen, Oxidants, Mercury and Aerosol Distributions, Sources and Sinks (NOMADSS) field campaign data (June 2013), which allows for direct comparison of isoprene mixing ratios with observations. The comparison between the modeled data and aircraft observations shows that isoprene mixing ratios agree well with measured isoprene for the M2.04 simulation but are overpredicted considerably by the M2.10 simulation.



# 1 Introduction

Biogenic emissions of volatile organic compounds play a fundamental role in atmospheric chemistry, specifically in the ozone cycle and in the formation of secondary organic aerosols with implications in air quality and climate. The major biogenic

volatile organic compounds (BVOCs) are isoprene and monoterpenes (e.g., α, β-pinene) with relative contributions of 69.2 %, and 10.9 %, respectively (Sindelarova et al., 2014). Emissions of BVOC have implications for air quality by affecting the concentration of ground-level ozone (Fehsenfeld et al., 1992; Curci et al., 2010; Sartelet et al., 2012; Sindelarova et al., 2014), and on climate through tropospheric ozone radiative forcing (Brasseur et al., 1998; Gauss et al., 2006). Churkina et al. (2017) estimated that the impact of BVOC emissions on ground level ozone production was on average 12 % in summer and up to 60

% during a heat wave event in the Berlin-Brandenburg metropolitan area, Germany (Churkina et al., 2017). With climate change, the increase of isoprene emissions from vegetation due to higher temperatures may lead to higher tropospheric ozone concentrations (EEA, 2015). In addition to the consequences in the gas-phase chemistry, oxidative products of some BVOCs can form secondary organic aerosol (Limbeck et al., 2003; van Donkelaar et al., 2007) with significant effects on the Earth's radiation budget.

The proper quantification of BVOC emitted into the atmosphere is a fundamental parameter in order to represent their effect reliably in global and regional chemical transport models (CTM). Therefore, several modelling approaches have been developed for the estimation of BVOC emissions (Guenther et al., 1995; Niinemets et al., 1999; Martin et al., 2000; Arneth et al., 2007). A fundamental step towards BVOC modelling relates to the work by Guenther et al. (2006) (G06 hereafter), who developed the Model of Emissions of Gases and Aerosols from Nature version 2.0 (MEGAN v2.0) for both regional and global

BVOC emission modelling. This model estimates the emissions considering meteorology (e.g., temperature, and solar radiation), leaf area index (LAI), and plant functional type (PFT) as driving variables, with higher emissions occurring for higher temperature, transmission of photosynthetic photon flux density, and LAI. MEGAN v2.0 was used for analyzing the impact of biogenic emissions with potential future increases in ambient temperature on ozone levels (Im et al., 2011), aerosol levels and chemical compositions (Im et al., 2012). Building on MEGAN v2.0 (G06) and MEGAN v2.02 (Sakulyanontvittaya

et al., 2008), Guenther et al. (2012) (G12 hereafter) introduced additional compounds, emission types, and controlling processes with MEGAN v2.1. In MEGAN v2.1, the emission factors are adjusted to consider that the measured net flux of BVOC compounds above the vegetation canopy does not involve the dry deposition flux, so that the net primary emissions would be higher (e.g., up to a few percent for isoprene). To better depict the variability of isoprene emission within a PFT category, MEGAN v2.1 allows specific PFT emission factors for each vegetation type.

Several global and regional-scale chemistry transport models have adopted MEGAN as their BVOC emission model, including the Weather Research and Forecast model coupled with chemistry (WRF-Chem - Grell et al., 2005; Fast et al., 2006). Zhao et al. (2016) investigated the sensitivity of WRF-Chem simulated BVOC emissions with different land surface schemes, namely the Community Land Model version 4.0 (CLM4 - Oleson et al., 2010; Lawrence et al., 2011) and the Noah land surface model (Niu et al., 2011), for MEGAN v2.0, and considering also different vegetation maps for MEGAN v2.1 implemented into the





CLM4. These authors found that BVOC emissions modelled with MEGAN v2.0 were negligible between the two runs with
different land surface schemes and one type of vegetation map, whereas considering the same land surface scheme with
different vegetation maps induced consistent variations in BVOC emissions predicted with MEGAN v2.1. Henrot et al. (2017)
implemented MEGAN v2.1 in ECHAM-HAMMOZ (ECHAM6 atmospheric general circulation model; HAM aerosol
model; MOZART chemistry transport model). These authors found values of isoprene and other compounds at the regional
scale similar to the findings from previous studies (Sindelarova et al., 2014; Messina et al., 2016) that used different
meteorological drivers, thus confirming that the emission factor and PFT distributions determine the spatial emission
distribution in MEGAN.  Jiang et al. (2019) utilized the WRF-CAMx (WRF meteorology model; CAMx regional air quality
model) modelling package to investigate the effect of BVOC emissions on the surface ozone levels in Europe. They found
higher (about 3 times) isoprene emissions predicted with MEGAN v2.1 compared to another BVOC emission model (i.e., Paul
Scherrer Institute model - Andreani-aksoyoglu and Keller, 1995) resulting in about 10 % higher ozone mixing ratios. Therefore,
Jiang et al. (2019) suggested that ozone production occurs generally in VOC-saturated rather than VOC-sensitive regimes in
Europe. A few species dominate the total isoprene and monoterpene emissions in European forests, with three Quercus species
and five species contributing to 66 % and 80 % of total isoprene and monoterpene emissions, respectively (Keenan et al.,
2009). Although Zhao et al. (2016) implemented MEGAN v2.1 in WRF-Chem with the CLM4 land model, it did not become
part of the community version of WRF-Chem until spring 2021 with the release of WRF version 4.3; the CLM surface scheme
and associated subroutines in the physics and chemistry packages have been modified to be consistent with the MEGANv2.1
biogenic emission. Here, we explore the effect of making simple changes to the existing WRF-Chem MEGAN v2.04 emissions
scheme in WRF-Chem to provide MEGAN updates that can be used independently of land surface model chosen. In section
2, we describe the changes that were made to MEGAN v2.04.

In order to compare different updates to MEGAN v2.0 introduced by Guenther et al. (2012) with MEGAN v2.1 in simulating
BVOC emissions, two case studies were performed in two different domains (i.e., Europe and the Southeast United States).
Since ozone is known to be a result of photochemistry involving nitrogen oxides ($NO_X = NO + NO_2$) and volatile organic
compounds (VOC), a sensitivity study on BVOC emissions was performed for a high-ozone episode in August 2015 in Europe
considering different updates to MEGAN v2.0 introduced with MEGAN v2.1. For this case study, comparisons are presented
between modelled ozone concentrations and surface measurements (AirBase database). Summer 2015 was among the six
hottest and driest summers since 1950 in Europe (Ionita et al., 2017). These meteorological conditions together with 4 heatwave
episodes led to high tropospheric ozone levels throughout Europe, with 18 of the EU-28 countries exceeding the EU ozone
threshold value for the protection of human health (EEA, 2017). Lin et al. (2020) have reported a link between ozone episodes
in Europe and the ecosystem-atmosphere interactions during heatwaves and droughts, with lower ozone uptake by water-
stressed vegetation exacerbating the peak ozone events. For the Southeast United States case study, BVOC emissions
calculated with MEGAN v2.0 and MEGAN v2.1 were evaluated against aircraft measurements. The measurements of isoprene,
two products of isoprene oxidation (i.e., methacrolein, and methyl vinyl ketone) and ozone were taken in five of the research
flights under the Southern Oxidant and Aerosol study (SOAS) in June 2013. The SOAS project is part of the Nitrogen,





Oxidants, Mercury and Aerosol Distributions, Sources and Sinks (NOMADSS) project
(https://www.eol.ucar.edu/field_projects/nomadss) under the umbrella of Southeast Atmosphere Study (SAS), a project aimed
at investigating the interactions between atmosphere and biosphere and the role of BVOC in atmospheric chemistry in the
Southeast and Central United States. A synthesis of relevant results achieved within SAS was presented by Carlton et al.
(2014). Section 3 describes these two cases in more detail and the WRF-Chem v3.9 configurations to represent the two cases.
In section 4, the effects of specific updates to the MEGAN v2.04 model are examined and evaluated with observations from
each of the case studies. A summary and conclusions are given in section 5.

## 2 Materials and Methods

### 2.1 Updates to MEGAN v2.04 in WRF-Chem

The Model of Emission of Gases and Aerosols from Nature (MEGAN) model estimates the net emission rate of 134 chemicals
species (e.g. isoprene, monoterpenes, oxygenated compounds, sesquiterpenes and nitrogen oxide) from terrestrial ecosystems
into the above-canopy atmosphere with a resolution of 1 $km^2$ (G06). MEGAN can be used in both global models, such as
GEOS-Chem (Goddard Earth Observing System) (Bey et al., 2001) or CAM-Chem (Community Atmosphere Model) (Tilmes
et al., 2015; Lamarque et al., 2012), and regional CTM, such as WRF-Chem (Grell et al., 2005; Fast et al., 2006).
The BVOC emission algorithm currently applied to WRF-Chem is calculated as follows:

$$EM \ = \ \varepsilon \cdot \gamma_{LAI} \cdot \gamma_P \cdot \gamma_T \cdot \gamma_{age} \cdot \gamma_{SM} \cdot \rho, \tag{1}$$

where EM is the BVOC emission rate ($\mu g \ m^{-2} \ hr^{-1}$); $\varepsilon$ the emission factor ($\mu g \ m^{-2} \ hr^{-1}$); $\gamma_{age}$, $\gamma_{SM}$, $\gamma_{LAI}$, $\gamma_P$ and $\gamma_T$ are the emission
activity factors that account respectively for leaf age, soil moisture, leaf area index, photosynthetic photon flux density and
temperature (normalized ratio); and $\rho$ the loss and production within plant canopy (normalized ratio) (G06). The emission rate
(EM) is calculated for each plant functional type (PFT), added up to estimate the total emission at each model grid cell, and
corrected taking into account the deviation from the standard condition ($\gamma$ and $\rho$ parameters).
In the present study, the changes made to the MEGAN algorithm implemented in WRF-Chem were the following: (i) update
of the emission activity factors ($\gamma_i$), (ii) update of emission factor values for each plant functional type (PFT), and (iii) the
assignment of the emission factor by PFT to isoprene.

### 2.2 Update of the emission activity factors

Emission activity factors describe variations in BVOC emission related to physiological and phenological processes. The
capability of a leaf to emit isoprene depends on a number of physical and biological factors, with incident photosynthetic
photon flux density (PPFD) and leaf temperature as driving factors (Guenther et al., 1993). A leaf's capacity to emit isoprene
is also influenced by leaf phenology, with young leaves emitting no isoprene and mature leaves emitting isoprene maximally.





Moreover, soil characteristics play a role in the plants BVOC emission ability, with droughts significantly influencing isoprene emission (Guenther et al., 2006; Jiang et al., 2018).

The integration of MEGAN in CTMs (e.g., temperature, solar radiation and soil moisture) allows examinations of interactions
between BVOC emissions, the surrounding environment and the canopy itself. The standard MEGAN environment model is based on the methods described by Guenther et al. (1999) who estimated incident PPFD and temperature at five canopy depths, including a leaf isoprene-emitting model driven by humidity, solar radiation, ambient temperature, and soil moisture. Overall, the BVOC emissions is a product of both the local state (temperature and PPFD) and the "climate" (soil moisture and heat waves/drought), hence the emissions are a function of both the instantaneous temperature and the temperature averaged over
1–10 days. Several algorithms have been widely used to simulate the response of isoprene emission to changes in light, temperature, leaf age and soil moisture (Guenther et al., 1995, 1999, 1993). However, complexity and expensive computational costs hindered their use in CTMs. To minimize computational costs, Guenther et al. (2006) developed a parameterized canopy environment emission activity (PCEEA) algorithm as an alternative to calculating all variables at each canopy depth and included it in the light, temperature and canopy environment response emission activity factors in MEGAN v2.04.

Note that this work simply replaces equations in the MEGAN v2.04 code with the equations in MEGAN v2.10. Table 1 lists the equations from MEGAN 2.04 with what they were replaced with from the MEGAN v2.10 paper (G12). One difference between this work and that of Guenther et al. (2012) is that this paper retains four plant functional types while Guenther et al. (2012) use 15 plant functional types. Details on the update of emission factors for this paper are given in section 2.3.

### 2.2.1 Light response emission activity factor

One of the main advances introduced with MEGAN v2.10 is that the emission activity factors of each compound class comprise are comprised of a light-dependent fraction (LDF) and a light-independent fraction (LIF). MEGAN v2.04 calculates the light response emission activity ($\gamma_p$) using the sine of the solar angle with no distinction between the light dependent and independent fractions (Eq. from (10) to (13) of Guenther et al. (2006)). For each class compound, the updated emission activity factor accounting for the PPFD variations is changed to the following equations:

$$\gamma_{P,i} = (1 - LDF_i) + LDF_i \cdot \gamma_{P\_LDF}, \tag{2}$$

$$\gamma_{P_{LDF}} = C_P \left[ \frac{\alpha \cdot PPFD}{(1 + \alpha^2 \cdot PPFD^2)^{0.5}} \right], \tag{3}$$

$$C_P = 0.0468 \cdot e^{(0.0005 \times [P_{24} - P_s])} \cdot [P_{240}]^{0.6}, \tag{4}$$

$$\alpha = 0.004 - 0.0005 \cdot \ln(P_{240}), \tag{5}$$

where the PPFD is the instantaneous photosynthetic photon flux density (µmol m$^{-2}$ s$^{-1}$); the Ps the standard conditions for PPFD averaged over the past day (200 µmol m$^{-2}$ s$^{-1}$ for sun leaves and 50 µmol m$^{-2}$ s$^{-1}$ for shade leaves); $P_{24}$ is the average PPFD of the past 24 hours; the $P_{240}$ is the average PPFD of the past 10 days (Table 1). This new version code calculates the $\gamma_p$ with the photosynthetic photon flux density using the internal variable "swdown": the downward solar radiation (W m$^{-2}$). $P_{24}$ and $P_{240}$





are the average PPFD of the past day and the past ten days, nevertheless, in the modified code, they are both equal to
"mswdown" variable: the downward solar radiation (W m$^{-2}$) of previous month (G12).

**2.2.2 Temperature response emission activity factor**

In MEGAN v2.04, the temperature activity factor ($\gamma_T$) calculates the response emission activity for isoprene according to Eq.
(5), Eq. (8), and Eq. (15) by Guenther et al. (2006); all the others non-isoprenoid compounds are described according to the
monoterpene exponential temperature response function by Guenther et al. (1993).

The updated temperature activity factor (MEGAN v2.10) leads to two different changes, (i) the introduction of LDF and LIF
(i.e., as the previous emission factor), and (ii) the dependency on the specific classes compounds instead of the isoprene and
non-isoprene species. The updated version of LDF of temperature activity factor ($\gamma_T$) is calculated as follows:

$$\gamma_{T,i} = (1 - LDF_i) \cdot \gamma_{T\_LIF,i} + LDF_i \cdot \gamma_{T\_LDF,i}, \qquad (6)$$

$$\gamma_{T_{LDF,i}} = E_{opt} \cdot \left[ CT_2 \cdot \frac{e^{(CT_{1,i} \cdot x)}}{CT_2 - CT_{1,i} \cdot (1 - e^{(CT_2 \cdot x)})} \right], \qquad (7)$$

$$x = \frac{\left[\left(\frac{1}{T_{opt}}\right) - \left(\frac{1}{T}\right)\right]}{0.00831}, \qquad (8)$$

$$E_{opt} = C_{eo,i} \cdot e^{(0.05 \cdot (T_{24} - T_S))} \cdot e^{(0.05 \cdot (T_{240} - T_S))}, \qquad (9)$$

$$T_{opt} = 313 + \left(0.6 \cdot (T_{240} - 297)\right), \qquad (10)$$

where $E_{opt}$ is the maximum normalized emission capacity (mol km$^{-2}$ hr$^{-1}$); $T_{opt}$ the temperature at which $E_{opt}$ occurs (K); T is
the leaf temperature (K); $CT_{1-i}$, $CT_2$, and $C_{eo-i}$ are emission-class dependent empirical coefficients; $T_s$ the standard conditions
for leaf temperature (297 K); $T_{24}$ the average leaf temperature of the past 24 hours (K); $T_{240}$ the average leaf temperature of
the past 240 hours (K).

The response of LIF is determined according to the monoterpene exponential temperature response function by Guenther et
al. (1993):

$$\gamma_{T_{LIF,i}} = e^{(\beta_i(T - T_S))}, \qquad (11)$$

where $\beta_i$ is an empirically determined coefficient, depending on the emission class compound (G12).

Additional changes made on this part of the code concern the update of the $CT_1$, $CT_2$ and $C_{eo}$ parameters. In Guenther et al.
(2006) their values are set respectively to 80, 200 and 1.75, whereas $CT_{1,i}$ and $C_{eo,i}$ depend on the classes compound, and $CT_2$
still have a fixed value (i.e., 230) in the updated version (Table 1). A more accurate BVOC evaluation with each class
compound having the appropriate value may result from (i) the temperature activity factor defined as the weighted average of
a light dependent and independent fraction ($\gamma_{Ti,LDF}$ and $\gamma_{Ti,LIF}$), (ii) and the update of the model parameters ($CT_1$, $CT_2$, and $C_{eo}$),
for each class compound. Note that the value of $T_{24}$ and $T_{240}$ is estimated equal to the variable monthly surface air temperature
(MTSA) with MEGAN v2.10. Therefore, it is assumed that the average temperature of the past 24 hours, and the past ten days,
are the same as the average temperature of the past month ($T_{24} = T_{240} = MTSA$).





### 2.2.3 Leaf age response emission activity factor

The canopy isoprene-emitting capability is also influenced by the leaf age. An increase in foliage is assumed to imply a higher
production of isoprene (young leaves), whereas decreasing foliage is associated to less production of isoprene (old leaves).
Guenther et al. (1999) developed an algorithm, with a time step of one month, to simulate the emissions change for young,
mature, and old leaves. The algorithm adapted to MEGAN v2.04 assumes a constant value ($\gamma_{age}$=1) for evergreen canopies,
while deciduous canopies are divided into four fractions: new foliage ($F_{new}$), growing foliage ($F_{gro}$), mature foliage ($F_{mat}$) and
old foliage ($F_{old}$). The leaf age factor is computed as

$$\gamma_{age} = F_{new}A_{new} + F_{gro}A_{gro} + F_{mat}A_{mat} + F_{old}A_{old}, \tag{12}$$

where $A_{new}$, $A_{gro}$, $A_{mat}$, and $A_{old}$ are the relative emission rates assigned to each canopy fraction depending on PFT categories.
The canopy is divided into leaf age fractions based on the change in LAI between the current time step (current month = $LAI_c$)
and the previous time step (previous month = $LAI_p$). The difference between the two LAI values describes the leaf area index
age. No difference in LAI (i.e., $LAI_p$=$LAI_c$) indicates a canopy mostly formed by mature foliage. A canopy is formed by old
foliage when the LAI value of previous month is greater than the one in the current month ($LAI_p$>$LAI_c$), whereas $LAI_p$<$LAI_c$
for a canopy primarily formed by new foliage (G06).

MEGAN v2.10 estimates the leaf age emission activity factor ($\gamma_{age}$) in Eq. (12) based on the same calculations described by
Eq. (16) in Guenther et al. (2006). The two versions of MEGAN do not differ for the canopy subdivision into four fractions
(i.e., new foliage ($F_{new}$), growing foliage ($F_{gro}$), mature foliage ($F_{mat}$), and old foliage ($F_{old}$)) and the related computation. The
only update of equation parameters is the relative emission rates assigned to each compound class ($A_{new}$, $A_{gro}$, $A_{mat}$, and $A_{old}$)
reported in Table 4 of Guenther et al. (2012) (**Error! Reference source not found.**Table 1).

### 2.2.4 Soil moisture response emission activity factor

Different studies have shown that isoprene emission decreases when soil moisture drops below a threshold, and eventually
becomes insignificant, when plants are exposed to extended drought (Jiang et al., 2018; Pegoraro et al., 2004). In MEGAN
v2.04, the soil moisture activity factor ($\gamma_{SM}$) is set to 1.0 for both isoprene and no-isoprene classes compound. Therefore, the
soil moisture dependence is not involved into the BVOC emissions algorithm. In the present study, (in MEGAN v2.10 code
applied to WRF-Chem), isoprene emissions were evaluated according to the Eqns. (20-a), (20-b) and (20-c) described by
Guenther et al. (2006) as follows:

$$\gamma_{SM,isoprene} = 1 \qquad (\theta > \theta_1) \tag{1}$$

$$\gamma_{SM,isoprene} = \frac{\theta - \theta_W}{\Delta\theta_1} \quad (\theta_W < \theta < \theta_1) \tag{2}$$

$$\gamma_{SM,isoprene} = 0 \qquad (\theta < \theta_W) \tag{3}$$

$$\theta_1 = \theta_W + \Delta\theta_1 \tag{4}$$



where θ is soil moisture ($m^3$ $m^{-3}$); $\theta_w$ is the soil moisture threshold below which plants cannot extract water from soil (wilting point, $m^3$ $m^{-3}$); $\Delta\theta_1$ (=0.06) is an empirical parameter from Pegoraro et al. 2004. MEGAN uses a wilting point database that assigns different $\theta_w$ values for each soil type based on Table 2 of Chen and Dudhia (2001) (Table S1 of supplemental materials). Since for Guenther et al. (2012) the non-isoprenoid soil moisture dependence is not involved into the BVOC emissions algorithm, in the present study, the $\gamma_{SM}$ for non-isoprenoid compounds is still set to 1.0.

**2.2.5 Canopy environment response emission activity factor**

The emission response to leaf area index ($\gamma_{LAI}$) in MEGAN v2.04, calculates the response emission activity factor by Eq. (15) of G06. In MEGAN v2.10 the canopy environment coefficient has been simplified as follows:

$$\gamma_{LAI} = LAI \cdot C_{CE}, \tag{17}$$

where LAI ($m^2$ $m^{-2}$) is the leaf area index referred to the month of the simulation; $C_{ce}$ (=0.57) is a value dependent on the canopy environment model.

**2.3 Updates of PFTs and isoprene emission factors**

An important difference between MEGAN v2.04 and MEGAN v2.10 is the number of PFTs described and the associated isoprene emission factors. Only four PFTs are used in MEGAN v2.04, including Needleleaf Trees, Broadleaf Trees, Broadleaf Shrubs, and Grass and other. In contrast, MEGAN v2.10 includes 15 PFTs (Needleleaf Evergreen Temperate Tree, Needleleaf Evergreen Boreal Tree, Needleleaf Deciduous Boreal Tree, Broadleaf Evergreen Tropical Tree, Broadleaf Evergreen Temperate Tree, Broadleaf Deciduous Tropical Tree, Broadleaf Deciduous Temperate Tree, Broadleaf Deciduous Boreal Tree, Broadleaf Evergreen Temperate Shrub, Broadleaf Deciduous Temperate Shrub, Broadleaf Deciduous Boreal Shrub, Arctic C3 Grass, Cool C3 Grass, Warm C4 Grass and Crop). In order to explore the effect of the updated emission factors without revising the pre-processing code, we opted to apply a typical emission factor from Table 2 of Guenther et al. (2012) to the four PFTs currently in WRF-Chem. Table 2 shows the new emission factors ($\mu g$ $m^{-2}$ $hr^{-1}$) applied, for each type of plants with comparisons to the old values. The new isoprene emission factor decreased, for all PFT, except for herbaceous species (HB - Grass and other); at the bottom of Table 2 it is noticeable that carbon monoxide, the bidirectional, the stress and the other VOC decreased with new values, independently the PFT considered. For all the other classes compound, the new emission factors are larger than the previous emission factors.

The PFTs emission factors update does not change the isoprene emission, as its emission factors in MEGAN v2.04 implemented in WRF-Chem are estimated directly from the input database. Thus, a sensitivity simulation was performed with the isoprene emission factor evaluated according to the MEGAN emission algorithm Eq. (1), instead of the input database as outlined in section 3.



## 3. Case Study Descriptions and Model Configuration

### 3.1 European Case

#### 3.1.1 Characterization of Case from Observations

Summer 2015 was among the six hottest and driest summers since 1950 in Europe (Ionita et al., 2017). In this year, high

tropospheric ozone episodes were experienced throughout Europe, with 18 of the EU-28 countries as well as 41 % monitoring stations reporting the ozone maximum daily eight-hour mean above 120 μg m$^{-3}$ (=60.4 ppb; the current target value for ozone in Directive 2008/50/EC) on more than 25 days (EEA, 2017). Therefore, a 6-day high ozone period (10–16 August 2015) was selected to evaluate the impact of the changes in the MEGAN v2.04 scheme on isoprene emissions and ozone mixing ratios. The high ozone levels were confirmed by examining the summertime (May–September) hourly average ozone concentrations

measured at the air-quality monitoring stations in Marche region (Italy) (Figure 1-a and Figure 1-b), over a period of 3 years (from 2013 to 2015). The analysis results indicate that an extraordinary ozone peak event occurred in the time period 10–16 August 2015. In particular, on August 13$^{th}$ 2015 all the air quality stations reported the highest ozone daily eight-hour mean concentration value of the whole year (Figure 1-c).

#### 3.1.2 Model Configuration

The WRF-Chem model (simulation domain showed in the Figure 2) used initial and boundary conditions from the FNL (Final) Operational Global Analysis data (Ncep, 2000). These data are available every 6-hourly on a (1° x 1°) spatial grid. As summarized in

Table 3, the following physical schemes were used. The Morrison double-moment scheme was selected for the treatment of the microphysics processes (Morrison et al., 2009). The Rapid Radiative Transfer Model (RRTMG), for both shortwave and

longwave radiation is used; this allows to activate the aerosol direct radiative effect (Iacono et al., 2008), to represent scattering and absorption in the atmosphere. The Mellor-Yamada-Janjic (MYJ) parameterization was considered to describe the planetary boundary layer (Janjić, 1994). The unified Noah land-surface model was chosen to represent the land surface interaction (Chen et al., 1996). It includes soil temperature and moisture in four layers, fractional snow cover and frozen soil physics. The Grell-Freitas scheme was considered for the cumulus parameterization scheme: it tries to smooth the transition to cloud-resolving

scales (Grell and Freitas, 2014).

To investigate the role of isoprene on the high ozone event recorded in Europe, the selected chemical package was the chemical option with the Model for Ozone and Related chemical Tracers (MOZART) version 4 (Emmons et al., 2010) for the trace gases, and the Model for Simulating Aerosol Interactions and Chemistry (MOSAIC) (Zaveri et al., 2008) for the aerosol-phase species. The CAM-chem (Tilmes et al., 2015; Lamarque et al., 2012) global model results are used for the chemical initial and

boundary conditions for both the gas and aerosol components. The Emission Database for Global Atmospheric Research-Hemispheric Transport of Air Pollution (EDGAR-HTAP) emission inventory for Europe provided the anthropogenic





emissions (Janssens-Maenhout et al., 2012). The open biomass burning emissions were from the Fire Inventory from NCAR-FINN model (Wiedinmyer et al., 2011) and the biogenic emissions from the MEGAN database (Guenther et al., 2006, 2012). Table 4 lists the four simulations conducted to study the MEGAN updates described above. The control run (M2.04) uses the

MEGAN v2.04 database without any changes. The second simulation (MG) includes only the changes to the activity factors (γ). The third simulation (MGPFT) adds the changes in the activity factors due to the variation of the PFTs emission factors. The fourth simulation (M2.10) is the same as the MGPFT run, except the isoprene emission factor is calculated with Eq. (1) instead of prescribed by the input data.

### 3.2 Southeast US Case

#### 3.2.1 Characterization of Case from Observations

The NOMADSS project integrates the objectives of three National Science Foundation (NSF) funded projects: the quantification of biogenic emissions and their interactions with anthropogenic pollutants (SOAS), the distribution and the chemical transformations of speciated mercury in the troposphere (NAAMEX) and the investigation of the role of particulate nitrate photolysis in the cycling of reactive nitrogen species in the troposphere (TROPHONO). NOMADSS took place over

the southeastern United States from June 1$^{st}$ to July 15$^{th}$, 2013. The NSF/NCAR C-130 flight tracks covered much of the eastern United States under the NOMADSS project with the aim of quantifying fluxes, transformations and distributions of VOC, ozone, NOx, mercury and HONO (nitrous acid) (https://www.eol.ucar.edu/field_projects/nomadss) (NOMADSS - Operational plan, 2013). The NOMADSS field campaign includes 19 flights from June 3$^{rd}$ to July 14$^{th}$, 2013. From those flights, the first five (rf01-rf05) (Figure 3), conducted mostly in the Southeast US to complement the SOAS objectives, were

selected to estimate and evaluate the updates of the MEGAN code within the WRF-Chem model version 3.9. The Figure 3 show that the aircraft sampled air in isoprene-rich emissions regions of United States, that the flights tracks show high isoprene mixing ratios when the aircraft was in the boundary layer, and therefore, the low isoprene mixing ratios occurred when the aircraft was in the free troposphere.

#### 3.2.2 Model Configuration

Figure 4 shows the model domains. The coarse domain has 442×265 grid points with 12 km grid cells centered at 40° N 97° W, covering the United States of America (USA) (CONUS domain – SW corner 22.83° N 120.49° W; NW corner 52.46° N 136.45° W; NE corner 45.98° N 60.82° E; SE corner 20.08° N 81.24 W). The nested domain is centered over the southeastern area of the USA with 301×301 grid points and 4 km grid cells including the selected NOMADSS flight tracks (rf01 - rf05) inside the simulation domain. Both domains consider 40 vertical levels up to 50 hPa. The simulations lasted 14 days, from

June 1$^{st}$ (00:00 UTC) to June 15$^{th}$ (00:00 UTC), 2013. The simulations started two days before the first flight (rf01 - June 3$^{rd}$) so as to guarantee a spin up for the model. To compare directly with the aircraft measurements, the "tracking" option was





selected in WRF-Chem. This option outputs the vertical profiles of prescribed meteorological and chemical species at a set of prescribed times and horizontal coordinates, taken from the location and time of the aircraft.

Meteorological boundary and initial conditions were extracted from NCEP North American Regional Reanalysis (NARR – ds608.0). The NARR project is an addition to the NCEP global reanalysis which is run only over the North American Region with the 32-km grid spacing of NCEP Eta model (NCAR, 2005). The configuration of the physical and chemical/aerosol schemes used for this part of the study is the same as that described in the previous section and reported in

Table 3. Two simulations were performed to evaluate the MEGAN model updates with the measurements sampled by the NCAR C-130. The two simulations were M2.04, with the original MEGAN v2.04 database, and M2.10 with all the code updates described previously in sections 2.2 and 3.1.

## 4. Results

### 4.1 European Case Study

In this section, we begin by describing and evaluating the synoptic meteorological conditions for August 10-15, 2015, as well as evaluating WRF-Chem temperature predictions with ground-based measurements, because isoprene emissions depend strongly on temperature. Then we show how isoprene and a-pinene emissions differ among the four simulations (M2.04, MG, MGPFT, and M2.10). Lastly, since BVOC observations are not available, trace gas ($NO_x$, CO and $O_3$) concentrations are compared between the different simulation concentration outputs with ground-based observations. The evaluation is conducted with both a statistical analysis based on the calculation of mean-bias, correlation coefficient and normalized root mean square error, and an assessment of the spatial distribution of the $NO_x$, CO and $O_3$ concentrations.

### 4.1.1 Synoptic conditions

We begin with evaluating the synoptic conditions predicted by the WRF-Chem simulations. The 6-day average geopotential height map at 850 hPa (Figure 5-a), shows the presence of an intense geopotential height maximum (1520–1580 m) affecting the central part of the Mediterranean basin. The ridge separates a geopotential height minimum (1300–1340 m) over north-western Europe from a weak depression (1460–1500 m) over Turkey. As a consequence, central and southern Europe are affected by north-easterly currents from north Europe, allowing the weak depression to cross Italy toward the southeast portion of the domain. The WRF-Chem (Figure 5-b), simulations are consistent with the evolution represented in the reanalysis, although with some slight differences. In particular, the low pressure is more intense in the WRF-Chem runs, whereas the high-pressure across Italy is more intense in the NCAR/NCEP reanalysis. The visual differences can be partially explained considering that the reanalysis with a coarse horizontal resolution of 2.5°×2.5° differs from the finer-resolution operational analysis used to force the model (12 km grids).

Comparison between simulated (Figure 5-d) and observed (Figure 5-c) 6-day average temperature shows that the values and the spatial distribution of temperature are well depicted by WRF-Chem model (Figure 5c-d). The lowest temperatures (i.e., 5–




10 °C) are in north-western Europe (i.e., Iceland). Temperatures increase in the north-easterly direction with values in the range of 10–15 °C in most parts of England and the Scandinavian Peninsula. Along the central and western Europe, the

temperature increases up to 15–25 °C (e.g., Portugal, Spain, French, Germany). Southeast Europe (e.g., Italy, Croatia, Albania, Greece, and Turkey) has the highest temperatures up to about 30–35 °C.

**4.1.2 Examination of the MEGAN emission algorithm updates**

The map of PFTs percentage coverage reveals higher coverage of Needleleaf trees compared to Broadleaf, and Shrub and Bush in north-eastern Europe with values between 30–70 % and, with more staggered trend, in the north of Spain (i.e., the Cantabrian

Mountains), Italy (i.e., Alps), Germany and in the most part of the Balkans peninsula (i.e., Carpathian Mountains) (Figure 6 a-d). The Broadleaf coverage has a geographical distribution similar to the Needleleaf trees, but with lower values (from 10 to 40 %). The shrub and bush PFT are predominant in Norway, north of Russia, south-eastern part of Spain, and Turkey. The Herbs cover the greatest portion of central Europe with a value 70-100 %, since there is a substantial number of plants that fall within this plant functional type (Grass and other - PFTP_HB). The isoprene emitting genera in this category include:

Phragmites (a reed), Carex (a sedge), Stipa (a grass) and Sphagnum (a moss) (G06).

Four European cities, Porto (Portugal), Genoa (Italy), Zagreb (Croatia), and Kiev (Ukraine) shown in Figure 2, were selected for analyzing the time series of isoprene and α-pinene emissions. These four cities represent warmer to cooler conditions experienced over Europe and are located in areas characterized by different PFTs (Figure 6). Figure 7 (a-d) shows the time series of isoprene emissions in the four selected cities from 10th to 16th August 2015. The isoprene diurnal cycle responds to

the daily fluctuations in solar radiation. The updates applied to MEGAN v2.04 in WRF-Chem resulted in increased isoprene emissions of up to 3 times for each city analyzed. Modifying the gamma factors (MG simulation) produced the greatest increase in emissions, while modifying the PFT emission factors with isoprene emission factors obtained from the input database (MGPFT) produced the same emissions magnitude as the MG simulation. Applying calculated isoprene emission factors (M2.10) gave lower isoprene emissions than MG and MGPFT, but still higher emissions than M2.04. The magnitude of the

isoprene emissions varied between cities where simulated isoprene emissions ranked as follows: Zagreb > Porto > Genoa > Kiev. Differences in the isoprene emission magnitudes are caused by the plant functional types in each city (Figure 6) and their respective emission factors. For example, Zagreb has about 30 %, 40 %, 20 %, and 70 % for BT, NT, SB, and HB vegetation, respectively, while Kiev has about 10 %, 30 %, 20 %, and 90 % for BT, NT, SB, and HB vegetation, respectively. Temperature and cloudiness can play a role in isoprene emission magnitude too. Figure 5 shows the temperature across Europe.

Porto has the temperature in the range of 20-25 °C, Zagreb 25-30 °C, Genoa 20-25 °C and Kiev 15-20 °C. Also, Porto and Genoa and possibly Kiev look like they may have experienced cloudiness based on the shape of the diurnal profile.

Figure 8 (a-d) shows the time series of α-pinene emissions for the selected cities from August 10th to August 16th, 2015. Among the monoterpene compounds, α-pinene is the highest contributor to the global annual BVOC emissions (Henrot et al., 2017). In each city, α-pinene emissions show daily patterns with peaks in the daytime and plateaus in the night-time, as with the

isoprene emissions but with an order of magnitude lower. In each of the city analyzed, the simulated α-pinene emissions ranked





as follows: MGPFT ≡ M2.10 > MG > M2.04. The α-pinene emissions from the MGPFT are the same as those from the M2.10 simulation, since the M2.10 code introduces only changes to isoprene emissions. As with the isoprene emissions, the updates to the gamma factors (MG) produced the greatest change in emissions, while modifying the emission factors (MGPFT) increased emissions somewhat more than the MG simulation. This result is consistent with the 10-20 % increase in emission

factors for NT and SB vegetation (Table 2). In general, the α-pinene emission values increase between 0.5 mol km$^{-2}$ hr$^{-1}$ (Kiev – 5$^{th}$ day) to 10 mol km$^{-2}$ hr$^{-1}$ (Porto – 1$^{st}$ day) compared to the control simulation (i.e., M2.04). As with isoprene, the differences in the isoprene emission magnitudes are caused by the plant functional types, temperature and cloudiness for each city.

Figure 9 and Figure 10, illustrate the spatial distribution of BVOC emissions calculated with different MEGAN configurations, respectively for isoprene and α-pinene emissions, as the weekly averaged emission flux (from August 10$^{th}$ 00:00 UTC to 16$^{th}$

00:00 UTC, 2015). The updates to the MEGAN algorithm introduce a significant increase in both isoprene and α-pinene emissions. The areas with higher increases (from 15 to 50 mol km$^{-2}$ hr$^{-1}$ for isoprene emissions; from 0.5 to 5 mol km$^{-2}$ hr$^{-1}$ for α-pinene emissions) in emissions are the Balkan Peninsula, the Apennine Mountains (Italy), and part of the Black Sea coasts (Turkey and Georgia). The Iberian Peninsula and central-east Europe show minor differences, but still noteworthy (from 15 to 35 mol km$^{-2}$ hr$^{-1}$ for isoprene; from 0.5 to 3.5 mol km$^{-2}$ hr$^{-1}$ for α-pinene). The increase of emissions on the Balkan Peninsula,

Italy, and Black Sea coast are likely a result of the substantial increase in γ$_{LAI}$ (Figure S3), which contrasts with the decreased Broadleaf PFT emission factor from 13000 to 9000 µg m$^{-2}$ hr$^{-1}$ (Table 2).

In Figure 11 (a-d), a comparison is presented between M2.04 run (green points) and the M2.10 (red points) emission activity factors γ$_P$, γ$_T$, γ$_{age}$, and γ$_{LAI}$ for the city of Genoa (Italy) in August 13$^{th}$ (12:00 UTC) 2015 (Figure S1, Figure S2 and Figure S3, of supplement material, show the remaining emission activity factors, respectively, for Kiev, Porto and Zagreb). The new

emission activity factors are substantially higher than those in MEGAN version 2.04. The PPFD gamma factor increases for isoprene from 1.25 in M2.04 to 2.3 in M2.10, which is 1.8 times greater, and for α-pinene from 1.0 to 1.2. While isoprene and other VOCs had little change in the temperature gamma factor (gamma_T); the γ$_T$ factor increased from 1.0 for M2.04 to 1.6 for M2.10. The leaf age emission activity factor (gamma_A) changed <10 % between M2.04 and M2.10, decreasing for isoprene and increasing for a-pinene. For all VOCs, the LAI gamma factor increased from 0.9 to 1.7, which has a substantial

effect on the VOC emissions. Figure 12 (a-d) shows the total emission activity factors (i.e., gamma = gamma_P*gamma_T* gamma_A*gamma_LAI) for each city between version 2.04 (M2.04, green points) and version 2.10 (M2.10, red points) of MEGAN equation for 12:00 UTC August 13, 2015. Compared to the M2.04 run, the emission activity values have increased significantly in the M2.10 run even considering the total value with an average value of about: 3 (Genoa and Zagreb), 0.6 (Porto), and 0.45 (Kiev). Naturally the increase of total values derives from the variation of single activity factors, for example

Genoa has seen its values double by updating the code from M2.04 to M2.10, particularly for γ$_P$, γ$_T$ and γ$_{LAI}$ (Figure 12), this attitude also applies to the city of Zagreb. The gap relative to Kiev and Porto instead, is mainly due to γ$_T$ and γ$_{LAI}$, although the PPFD activity factor (γ$_P$) seems has lower influence.



### 4.1.3 Evaluation of trace gas compounds

About 3000 air quality monitoring stations of 34 countries across Europe were analyzed from the AirBase database
(https://www.eea.europa.eu/data-and-maps/data/aqereporting-8) for $O_3$, and its precursors (i.e., CO, and $NO_2$). Since discrepancies between modelled and measured values might be related to the type and location of a monitoring station, the selected stations were also disaggregated into three class station types, namely urban, suburban and rural surface monitoring stations (Henne et al., 2010). For each station the weekly mean of the concentrations was calculated for the daytime hours, from 7:00 am to 18:00 pm UTC. The mean bias, the normalized root means square error, and the correlation coefficient were
calculated between the measured and simulated compounds (i.e., $O_3$, $NO_2$, and CO) for the different station classes (i.e., urban, suburban, and rural stations).

Regardless of the location of the monitoring stations, the M2.10, MG, MGPFT runs show similar statistics for ozone, with a consistent overestimation of ozone concentrations compared to the M2.04 run at each type of monitoring station. For each model run and type of station, comparison between modelled and measured ozone concentrations shows positive mean bias
values in the range 15–41% (Table 5). The ozone concentrations in rural areas present the lowest biases (M2.04 = 15 %, MG = MGPFT = 24 % and M2.10 = 23 %), while the highest biases are from the urban scenario (M2.04 = 31 %, M2.10 = 40 % and MG = MGPFT = 41 %). The MEGAN updates increase the mean biases of ozone concentrations by about 10 % regardless of the type of station considered. The changes to the MEGAN algorithm (i.e., MG, MGPFT, and M2.10 runs) have a small to negligible effect on modelled $NO_2$ and CO, with only CO having an increase of 2–4 % from the control simulation M2.04
compared to the other model runs (Table 5). Therefore, since changes to $NO_2$ and CO mixing ratios were small, the increase in the $O_3$ biases are most likely due to the increased biogenic VOC emissions.

There are strong positive correlations between modelled and observed $O_3$ concentrations, with slightly higher values of the correlation coefficient for MG, MGPFT, and M2.10 compared to M2.04. The ozone correlation coefficients are higher for the rural monitoring stations ($O_{3\text{-rural}}$ = 0.84-0.86), followed by the urban and suburban stations with values of about 0.75.
Comparisons between modelled and measured ozone concentrations at rural background monitoring stations limit the influence of the model resolution (Table 5) (Jiang et al., 2019). Table 5 presents nitrogen dioxide correlation coefficient values in the range of 0.22–0.43, again with the lowest values for the urban and suburban stations and, the correlation coefficients for CO, with low values (-0.02 to 0.22) for all the types of monitoring stations. There are no remarkable modifications with the different MEGAN updates simulations: $O_3$ and CO have an increase of about 0.01–0.02 from the control run (M2.04) to the MEGAN
updates simulations (MG, MGPFT, and M2.10); nitrogen dioxide coefficient correlation has no modifications. Figure 13 displays scatter plots and regression lines having on the ordinate axis the observed (AirBase dataset) concentrations, and on the abscissa axis, the simulations performed (i.e., M2.04 and M2.10 runs). The concentrations of $O_3$, $NO_2$, and CO observed and modelled support the statistical analysis of biases, RMSEs, and correlation coefficients; the high biases in $O_3$ tend to occur at locations where there are substantial discrepancies (up to 90 ug m$^{-3}$).





To learn how the spatial variation compares between observed and predicted trace gases, maps of mean day time (7 am – 6 pm UTC) concentrations of $O_3$, CO, and $NO_2$ (Figure 14) are examined for both the M2.04 control simulation and the M2.10 run with all the MEGAN code updates included. The spatial distribution of modelled ozone concentrations depicts well the observed values. However, the overestimation of the $O_3$ concentrations compared to the Airbase data is about 20 µg m$^{-3}$ (about 10 ppb) and up to about 40 µg m$^{-3}$ (about 20 ppb) for the M2.04 and M2.10 simulations, respectively. The overestimation is

visible for most of Europe, but it is more evident in central Europe (France, Germany, Switzerland, Austria and Northern Italy) and the south coast of the Iberian Peninsula. The results here contrast with those by Jiang et al. (2019), who found modelled ozone using the BVOC emission input from MEGAN v2.1 to be overestimated at low mixing ratios (20–50 ppb) and generally underestimated at mixing ratios above 50 ppb irrespective of the region of Europe considered. The $NO_2$ (Figure 14-b) spatial resolution is not well represented in WRF-Chem, especially in north Europe (i.e., England, Belgium, Netherlands and North

Germany), Northern Italy and Northeastern Spain. There is a large underestimation of $NO_2$ by the model in central Europe, where the difference is a factor of 10 (from 5 to 50 µg m$^{-3}$ - approximately from 2.5 to 5 ppb). This may be due to the lack of updated anthropogenic emissions as the EDGAR-HTAP emissions (Janssens-Maenhout et al., 2012) represent 2010 not 2015 which impacts the nitrogen oxides that are mainly emitted from anthropogenic sources (e.g. road traffic), or due to the 12-km grid spacing in WRF-Chem not resolving high concentrations in urban locations.

The WRF-Chem model underestimates CO concentrations by a factor of 2 (from 240 to 500 µg m$^{-3}$ - approximately from 210 to 435 ppb) for most of the stations measured (Figure 14-c), with the measured CO spatial distribution having no definite geographic pattern. The difference between measured and modelled CO concentrations is more evident across Italy, the south of Spain, Poland and Czech Republic. The magnitude of the gap in Eastern Europe could be a sign the model biomass burning emissions (FINN emissions - Wiedinmyer *et al.*, 2011) could not represent well the overview of the situation. Figure S4 shows

a comparison between weekly average CO concentrations evaluated with the M2.10 run (Figure S4-a), and a simulation with the same model setup with no biomass burning emissions (Figure S4-b - "M2.10_noFINN"). The difference between the two simulations is clear in Eastern Europe, without including the biomass burning emissions the CO concentration decreases from 240-320 to 160-240 µg m$^{-3}$ interval (209-280 to 140-209 ppb). This could indicate a presence of wildfire in that area, captured by both the Airbase dataset and the model, but probably, not well represented by biomass burning emissions and their

computation in WRF-Chem. Moreover, both the CO and $NO_2$ concentrations do not show differences in spatial resolution and concentration magnitude between the MEGAN update simulations (Figure 14).

**4.2 Southeast US Case Study**

Since the Southeast US Case Study encompasses the NOMADSS field campaign, simulated biogenic VOCs and other trace gases can be evaluated. The MEGAN code updates are compared with the NOMADSS NCAR C-130 flight measurements to

investigate the ability of the M2.04 and M2.10 simulations in depicting the BVOC composition in the boundary layer.





### 4.2.1 Evaluation of trace gas compounds

Figure 15 shows the altitude of the flight, the temperature, the concentrations of isoprene, methacrolein (MACR), methyl vinyl ketone (MVK), and ozone measured during the second flight (rf02 - Figure 15), of the NOMADSS field campaign, which
occurred on June 5th, 2013 (14 - 21 UTC; 9 - 16 US central daylight time). Isoprene, MACR, and MVK were measured by The Trace Organic Gas Analyzer (TOGA), which is a fast online Gas Chromatograph/Mass Spectrometer (GC/MS), with a measurement frequency of approximately one 30s sample every 2 minutes (Apel et al., 2003). Uncertainties of isoprene, MACR, and MVK are reported to be 15%, 20%, and 20% of the measured mixing ratio, respectively.

To explore the PBL ozone evolution, we examine the rf02 flight measurements since these measurements have a clear time
frame (Figure 15- a) (i.e., from 16:15 – 18:45 UTC time or 11:15 – 13:45 US central daylight time) when the flight track height was lower than the PBL height as it was flying near the Texas-Louisiana border (Figure 3). Comparison of M2.04 and M2.10 simulations to aircraft observations shows that isoprene (Figure 15-c) mixing ratios agree well with measured isoprene for the M2.04 simulation but are overpredicted by up to a factor of 5 in the PBL. In response, MACR (Figure 15-d), a product of isoprene (Table S2 in the supplement materials), is also overpredicted by the M2.10 simulation by up to a factor of 4, while
MACR is either well-predicted or overestimated by up to a factor of 2 by the M2.04 simulation. MVK (Figure 15-e), an isoprene-depended compound, has the opposite trend. That is MVK flight track measurements are more similar to the M2.10 run than the M2.04 simulation. Reasons in response to the higher isoprene, ozone mixing ratios (Figure 15-f) are affected with MEGAN v2.04 results showing more similarity to measurements than the M2.10 simulation, which generally overpredicts ozone by 10-20 ppbv. Commonly, CTMs tend to significantly overestimate surface ozone in the U.S. (Brown-Steiner et al.,
2015; Fiore et al., 2009; Lin et al., 2008). Recent studies have shed a light on modelling surface ozone in the southeast U.S. ( Travis et al., 2016; Schwantes et al., 2020; Cuchiara et al., 2020). Travis et al. (2016) investigated the main driving factors for the overestimation of modelled surface $O_3$ concentrations in the Southeast U.S. comparing CTM (i.e, Geos-Chem) predictions with multiplatform observations. These authors observed that a correction to the high-biased $NO_x$ emissions led to better matching modelled and measured $O_3$ concentrations both in the PBL and in the free troposphere. Cuchiara et al. (2020) have
investigated the interactions between cloud microphysics and the convective transport of soluble $O_3$ precursors from PBL to the upper troposphere. These authors applied a 50 % reduction to the biogenic isoprene emission calculated with MEGAN v.2.04 for WRF-Chem 3.9.1 based on the bias observed by previous studies in the U.S. southeast. A comprehensive study by Schwantes et al. (2020) dealt with a more detailed description of isoprene and terpene chemistry for modelling surface ozone with CAM-chem during the summer 2013 time period. Based on sensitivity tests, Schwantes et al. (2020) observed that the
more detailed isoprene chemistry representation improved agreement with the surface ozone daily max 8 h average values. Besides, a paper by Ryu et al. 2018 clarifies the effect of cloud prediction on ozone: having clouds in the right place at the right time also improved ozone predictions. Nevertheless, for our study this is likely not the cause, and the ozone overprediction is mainly due to the isoprene emission changes. According to large-eddy simulations (Kim et al., 2016; Li et al., 2016; Ouwersloot et al., 2011) and measurement-model analysis (Kaser et al., 2015) the effects of physical separation of isoprene





and OH in the PBL depends on chemistry-turbulence interactions and scale dependent heterogeneity of isoprene emissions, with potential implications on CTMs. The differences observed between measured and modelled isoprene mixing ratios along flight tracks may depend on the complex interaction between chemical reactions involving isoprene and turbulence within the PBL (Zhao et al., 2016). However, differences between modelled and aircraft data likely do not depend on boundary layer meteorological variables as measurements flights generally take place under weather conditions and boundary layer heights scarcely affected by boundary layer mixing phenomena (Travis et al., 2016).

## 5. Conclusions

To compare different updates to MEGAN v2.04 introduced by Guenther et al. (2012) with MEGAN v2.1 in simulating BVOC emissions, two case studies were performed in two different domains (i.e., Europe and the Southeast United States). A sensitivity study on BVOC emissions was performed for a high-ozone episode in August 2015 in Europe considering a control run with MEGAN v.2.04 (i.e., M2.04) and the (i) update of the emission activity factors (i.e., MG), (ii) update of the emission factor values for each plant functional type (PFT) (i.e., MGPFT), and (iii) the assignment of the emission factor by PFT to isoprene (i.e., M2.10).

Comparisons between modelled and surface measured (Airbase database) ozone concentrations showed values of the correlation coefficients in the range from 0.78 to 0.86, with higher values for the rural monitoring stations compared to the urban and suburban ones as well as for the M2.10 run compared to the M2.04 simulation. Moreover, the spatial distribution of modelled $O_3$ concentrations represented well the observed values, regardless of the simulations considered (M2.04, MG, MGPFT, and M2.10). However, magnitude differences were observed in both M2.04 and M2.10 simulations, with an overestimation of the $O_3$ concentrations compared to the Airbase data by about 20 μg m$^{-3}$ (10 ppb) and up to about 40 μg m$^{-3}$ (20 ppb), respectively.

For the Southeast United States case study, modelled BVOC emissions were evaluated against aircraft measurements to investigate the performance of M2.04 and M2.10 runs in depicting the BVOC dynamics in the planetary boundary layer (PBL). The measurements of isoprene, two products of isoprene oxidation (i.e., methacrolein, and methyl vinyl ketone) and ozone were taken in five of the research flights under the Southern Oxidant and Aerosol study in June 2013. To analyze the PBL ozone evolution, flight measurements when the flight track height was lower than the PBL height showed that the M2.04 simulation better represented the flight track isoprene mixing ratios than the M2.10 simulation. Each of the five research flights examined showed a M2.10 overestimation of isoprene mixing ratios up to a factor of 5. Comparisons between measured and modelled methacrolein and ozone reflected the isoprene comparison, with M2.04 results more similar to flight track measurements than the updated M2.10 simulation. Methyl vinyl ketone showed an opposite trend to the isoprene one, with the M2.10 results more like the flight track measurements than the control simulation.

In summary, the MEGAN updates (M2.10) generate substantially higher emissions of BVOCs, by factors of two or more. For both situations modeled here, better agreement with observations is obtained using the older emissions (M2.04). Mainly, the





simulations with updated emissions incurred larger biases in ozone measured across Europe, and overpredicted the concentrations of BVOC and their oxidation products observed directly during aircraft flights in the southeastern U.S. Both comparisons move us toward the assessment that BVOC emissions are better represented in M2.04 than in M2.10, although

of course the newer model has more flexibility to allow for future improvements. As explained before, we observed that the update to the algorithm which creates the major gap in the estimated BVOC emissions is the introduction of the revised emission activity factors ($\gamma_i$) (Figure 11 and Figure 12); they are substantially greater than the previous MEGAN version having an increment on average of about 0.5 and 1.5 considering, respectively, the individual and the total of factors. In order to study those differences, and make updates discussed in the present study as an consistent and stable improvement for the WRF-

Chem model, a next step could be the use and comparison of satellite observations of BVOC emissions directly (i.e., formaldehyde-HCHO, from the Ozone Monitoring Instrument-OMI) with the model updated for a long time frame simulation (i.e., month to year), and focusing on regional scale (i.e., whole Europe or United Station), as  Curci et al. 2010 have already performed. In conclusion, the updated code in this study (M2.10), to the detriment of what has been presented so far, is not ready yet to become part of the community version of WRF-Chem. As stated before, MEGAN v2.1 has already been

implemented (with the CLM4 land model) in the new release of WRF version 4.3. In fact, Zhao et al. (2016) modify the CLM surface scheme in order to be made consistent with the MEGANv2.1 biogenic emission. The main difference form this study and what Zhao et al. (2016) have done, is that we investigate the effect of changes to the existing WRF-Chem MEGAN v2.04 emissions scheme, and associated subroutines, in WRF-Chem to provide updates that can be used independently of land surface model used.

*Code availability.* The WRF-Chem code is available at [http://www2.mmm.ucar.edu/wrf/users/download/get_source.html](http://www2.mmm.ucar.edu/wrf/users/download/get_source.html). The code of updated MEGAN version (WRF-Chem modules: module_bioemi_megan2.F and module_data_megan2.F) can be obtained from Mauro Morichetti (m.morichetti@isac.cnr.it).

*Author contributions*. The first author (MM) developed and implemented the updates, performed the simulations and the analysis, and drafted the paper. MB contributed to simulation design, to the interpretation of results, and to writing the paper.
SM contributed to conceptualizing the paper. GP contributed to funding acquisition. EM and SV helped to review and editing the paper. UR supervised the entire writing-editing paper process.

*Competing interests.* The authors declare that they have no conflict of interest

*Acknowledgments.* Mauro Morichetti acknowledges funding from NCAR's Advanced Study Program's Graduate Student (GVP) fellowship. We would like to acknowledge high-performance computing support from Cheyenne
(doi:10.5065/D6RX99HX) provided by NCAR's Computational and Information Systems Laboratory, sponsored by the National Science Foundation. We also acknowledge use of the WRF-Chem preprocessor tool (mozbc, fire_emiss, megan_emission and antro_emission) provided by the Atmospheric Chemistry Observations and Modeling Lab (ACOM) of NCAR. We greatly appreciate the contributions from Christine Wiedinmeyer in providing advice in conducting the project. We greatly appreciate the colleagues who obtained the measurements used in the paper. For the NOMADSS aircraft data, we
acknowledge the NCAR-EOL-RAF team for state parameter data, A. Weinheimer, D. Knapp, D. Montzka, F. Flocke, and T. Campos for the ozone measurements, and E. Apel and R. Hornbrook for the volatile organic compounds data.



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



**Table 1: The emission activity factors equations referred to MEGAN version 2.04 (M2.04) and the relative updates made for version 2.10 (M2.10)**

| Emission Activity Factors | | M2.04* | M2.10* |
|---|---|---|---|
| Light response | | $$\gamma_P = 0 \qquad a < 0 \ a > 180$$ $$\gamma_P = sin(a)\left[2.46\left(1 + 005 \cdot (P_{daily} - 400)\right)\right)\varphi \cdot 0.9\ \varphi^2\right]$$ $$0 < a < 180$$ | $$\gamma_{P,i} = (1 - LDF_i) + LDF_i \cdot \gamma_{P\_LDF}$$ $$\gamma_{P_{LDF}} = C_P\left[\frac{\alpha \cdot PPFD}{(1 + \alpha^2 \cdot PPFD^2)^{0.5}}\right]$$ |
| Temperature response | Isoprene | $$\gamma_{T,isoprene} = \frac{E_{opt} \cdot CT_2 \cdot exp(CT_1 \cdot x)}{(CT_2 - CT_1 \cdot (1 - exp(CT_2 \cdot x)))}$$ Note, CT are fixed values | $$\gamma_{T,i} = (1 - LDF_i) \cdot \gamma_{T\_LIF,i} + LDF_i \cdot \gamma_{T\_LDF,i}$$ $$\gamma_{T_{LDF,i}} = E_{opt} \cdot \left[\frac{e^{(CT_{1,i} \cdot x)}}{CT_2 \cdot CT_2 - CT_{1,i} \cdot (1 - e^{(CT_2 \cdot x)})}\right]$$ $$\gamma_{T_{LIF,i}} = e^{(\beta_i(T - T_S))}$$ |
| | Non isoprene compounds | $$\gamma_T = e^{(\beta_i(T - T_S))}$$ | Note, CT values have been updated |
| Leaf age response | | $$\gamma_{age} = F_{new}A_{new} + F_{gro}A_{gro} + F_{mat}A_{mat} + F_{old}A_{old}$$ Anew, Agro, Amat, and Aold are fixed values | Anew, Agro, Amat, and Aold updated using Table 4 of G12 |
| Soil moisture response | Isoprene | $$\gamma_{SM} = 1$$ | $$\gamma_{SM,isoprene} = 1 \qquad (\theta > \theta_1)$$ $$\gamma_{SM,isoprene} = \frac{\theta - \theta_W}{\Delta\theta_1} \quad (\theta_W < \theta < \theta_1)$$ $$\gamma_{SM,isoprene} = 0 \qquad (\theta < \theta_W)$$ |
| | Non isoprene compounds | | $$\gamma_{SM} = 1$$ |
| Canopy environment response | | $$\gamma_{LAI} = \frac{0.49 LAI_C}{[(1 + 0.2LAI_C^2)^{0.5}]}$$ | $$\gamma_{LAI} = LAI \cdot C_{CE}$$ |

*See text for definitions of variables.




**Table 2: Biogenic emission classes and emission factors (new and old) (µg m⁻² hr⁻¹) for each plant functional types updated to MEGAN v2.10 applied to WRF-Chem (Guenther et al., 2012).**

|  | BT_2.04 | BT_2.10 | NT_2.04 | NT_2.10 | SB_2.04 | SB_2.10 | HB_2.04 | HB_2.10 |
|---|---|---|---|---|---|---|---|---|
| **Isoprene** | 13000 | 9000 | 2000 | 1800 | 11000 | 3333 | 400 | 866 |
| **Myrcene** | 20 | 50 | 75 | 70 | 22 | 36 | 0.3 | 0.3 |
| **Sabinene** | 45 | 62 | 70 | 70 | 50 | 56 | 0.7 | 0.7 |
| **Limonene** | 45 | 80 | 100 | 100 | 52 | 73 | 0.7 | 0.7 |
| **3-Carene** | 18 | 34 | 160 | 160 | 25 | 53 | 0.3 | 0.3 |
| **t-β-Ocimene** | 90 | 132 | 60 | 70 | 85 | 110 | 1 | 2 |
| **β-Pinene** | 90 | 126 | 300 | 300 | 100 | 116 | 1.5 | 1.5 |
| **α-Pinene** | 180 | 480 | 450 | 500 | 200 | 233 | 2 | 2 |
| **Other Monoterpenes** | 90 | 150 | 180 | 180 | 110 | 140 | 4.8 | 5 |
| **α-Farnesene** | 35 | 48 | 30 | 40 | 30 | 40 | 0.50 | 3 |
| **β-Caryophyllene** | 30 | 48 | 60 | 80 | 45 | 50 | 0.90 | 1 |
| **Other Sesquiterpenes** | 75 | 108 | 110 | 120 | 85 | 100 | 1.40 | 2 |
| **232-MBO** | 0.1 | 0.41 | 100 | 380 | 1 | 0.01 | 0.01 | 0.01 |
| **Methanol** | 800 | 740 | 800 | 900 | 800 | 900 | 800 | 500 |
| **Acetone** | 240 | 240 | 240 | 240 | 240 | 240 | 80 | 80 |
| **CO** | 1000 | 600 | 1000 | 600 | 1000 | 600 | 1000 | 600 |
| **Bidirectional VOC** | 1000 | 500 | 1000 | 500 | 1000 | 500 | 1000 | 80 |
| **Stress VOC** | 1000 | 280 | 1000 | 300 | 1000 | 300 | 1000 | 300 |
| **Other VOC** | 1000 | 140 | 1000 | 140 | 1000 | 140 | 1000 | 140 |



**Table 3: Namelist settings of the physical parameterizations used in the WRF-Chem setup simulations.**

| Model scheme | | Reference |
|---|---|---|
| **Microphysics** | Morrison two moment | (Morrison et al., 2009) |
| **Longwave radiation** | RRTMG | (Iacono et al., 2008) |
| **Shortwave radiation** | RRTMG | (Iacono et al., 2008) |
| **PBL model** | MYJ | (Janjić, 1994) |
| **Land surface** | Unified Noah land-surface | (Chen et al., 1996) |
| **Cumulus parameterization** | Grell-Freitas | (Grell and Freitas, 2014) |
| **Gas phase chemical mechanism** | MOZART version 4.0 | (Emmons et al., 2010) |
| **Aerosols representation** | 4-bin MOSAIC | (Zaveri et al., 2008) |


**Table 4: Simulations performed in this study.**

| Model run* | Emission activity factors ($\gamma_i$) modified | PFTs emission factors updated | Isoprene emission factor calculated by the MEGAN algorithm |
|---|---|---|---|
| **M2.04** | No | No | No |
| **MG** | Yes | No | No |
| **MGPFT** | Yes | Yes | No |
| **M2.10** | Yes | Yes | Yes |

*M2.04 = MEGAN v2.04, MG = only activity factors updated, MGPFT = activity factors and PFT emission factors updated, M2.10 = MGPFT plus including the update to the isoprene emission factor.






**Table 5: Summary of the statistics between predicted and measured ozone, NO₂ and CO concentrations from the Airbase dataset (intended as daytime hours weekly mean from August 10th, 2015 at 0000 UTC to August 16th, 2015 at 0000 UTC), namely the (a) normalized mean bias (bias - %), (b) normalized root mean square errors (nrmse – dimensionless), (c) the correlation coefficient (r - dimensionless), and the relative number of points analysed ($N_{XY}$). Values are shown according to the different station areas: suburban (SUB), urban (URB) and rural (RUR), and the different WRF-Chem model updates (control simulation - M2.04, activity factors updated - MG, PFTs emission factors updated - MGPFT, and the isoprene emission factor updated - M2.10).**

|  | $O_3$ | | | $NO_2$ | | | CO | | |
|---|---|---|---|---|---|---|---|---|---|
|  | **SUB** | **URB** | **RUR** | **SUB** | **URB** | **RUR** | **SUB** | **URB** | **RUR** |
| **$N_{XY}$** | 515 | 891 | 576 | 592 | 1602 | 487 | 151 | 637 | 73 |
| **bias_M2.04** | 26 | 31 | 15 | -39 | -63 | -3 | -27 | -35 | -29 |
| **bias_MG** | 37 | 41 | 24 | -39 | -63 | -2 | -23 | -31 | -26 |
| **bias_MGPFT** | 37 | 41 | 24 | -39 | -63 | -3 | -23 | -31 | -26 |
| **bias_M2.10** | 36 | 41 | 23 | -39 | -63 | -2 | -23 | -32 | -26 |
| **nrmse_ M2.04** | 0.34 | 0.39 | 0.23 | 0.82 | 0.96 | 0.89 | 0.71 | 0.80 | 0.71 |
| **nrmse_MG** | 0.43 | 0.49 | 0.30 | 0.82 | 0.96 | 0.90 | 0.70 | 0.78 | 0.70 |
| **nrmse_ MGPFT** | 0.43 | 0.49 | 0.30 | 0.82 | 0.96 | 0.90 | 0.70 | 0.78 | 0.70 |
| **nrmse_ M2.10** | 0.42 | 0.48 | 0.29 | 0.81 | 0.96 | 0.90 | 0.70 | 0.78 | 0.70 |
| **r_ M2.04** | 0.75 | 0.75 | 0.84 | 0.24 | 0.22 | 0.43 | 0.11 | 0.20 | -0.02 |
| **r_MG** | 0.77 | 0.76 | 0.85 | 0.24 | 0.22 | 0.43 | 0.12 | 0.22 | 0.00 |
| **r_ MGPFT** | 0.77 | 0.76 | 0.85 | 0.24 | 0.22 | 0.43 | 0.12 | 0.22 | 0.00 |
| **r_ M2.10** | 0.78 | 0.76 | 0.86 | 0.24 | 0.22 | 0.43 | 0.12 | 0.21 | 0.00 |








**Figure 1: (a-b) Marche region (Italy) air quality monitoring stations analysed in the 3 years study. (c) The ozone maximum daily eight-hour mean (ug m⁻³), averaged over all stations (b), in the month of August of 2013, 2014 and 2015 (http://94.88.42.232:16382 - ARPAM).**






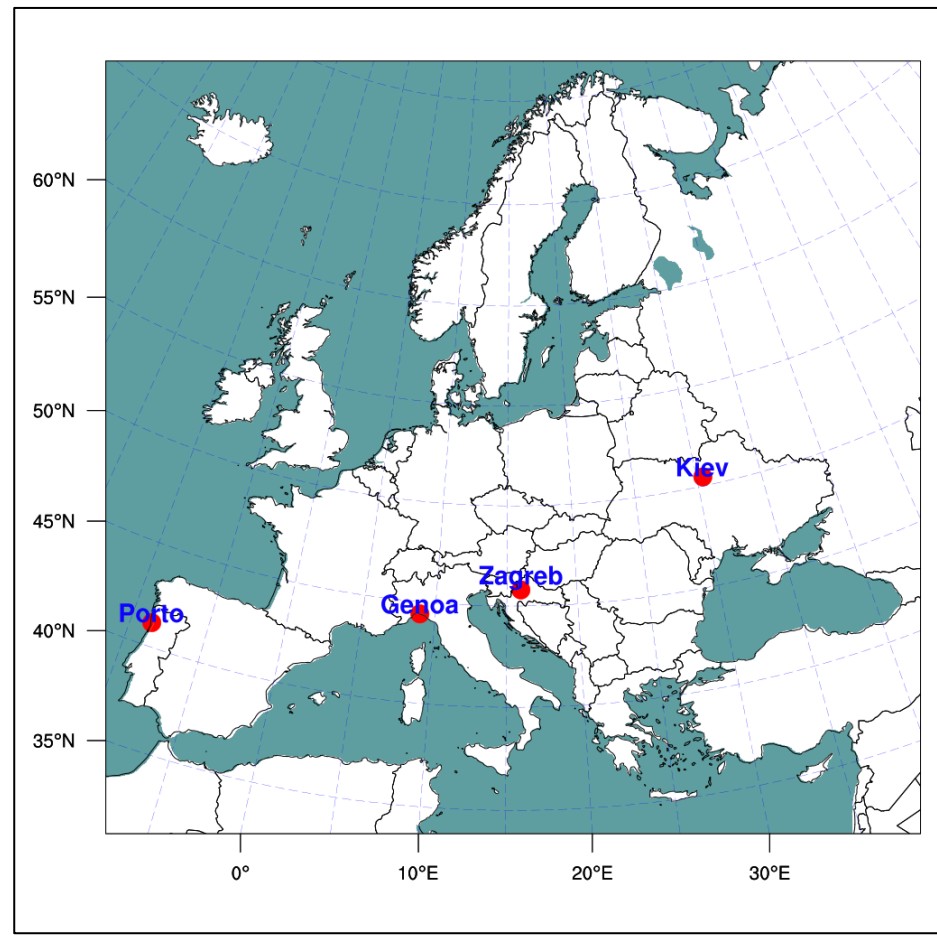

**Figure 2: The numerical domain of the WRF-Chem simulations with 380×360 grid points and 12 km grid cells and the location of the four cities in Europe selected for analyzing the simulated isoprene and α-pinene emissions, namely Porto (Portugal), Genoa, (Italy), Zagreb, (Croatia), and Kiev, (Ukraine) spanning in the range 41.15-51.45 °N and 8.63° W-30.50° E.**






**Figure 3: Flight tracks and the relative aircraft-based measurements of isoprene concentrations (pptv) under the Southern Oxidant and Aerosol Study plotted over the different maps of isoprene emission factors (mol km$^{-2}$ h$^{-1}$) from WRF-Chem output, for each day of the research flight at 3:00 pm local time (20:00 UTC), namely (a) rf01: 03/6/13; (b) rf02: 05/6/13; (c) rf03: 08/6/13; (d) rf04: 12/6/13; (e) rf05: 14/6/13.**




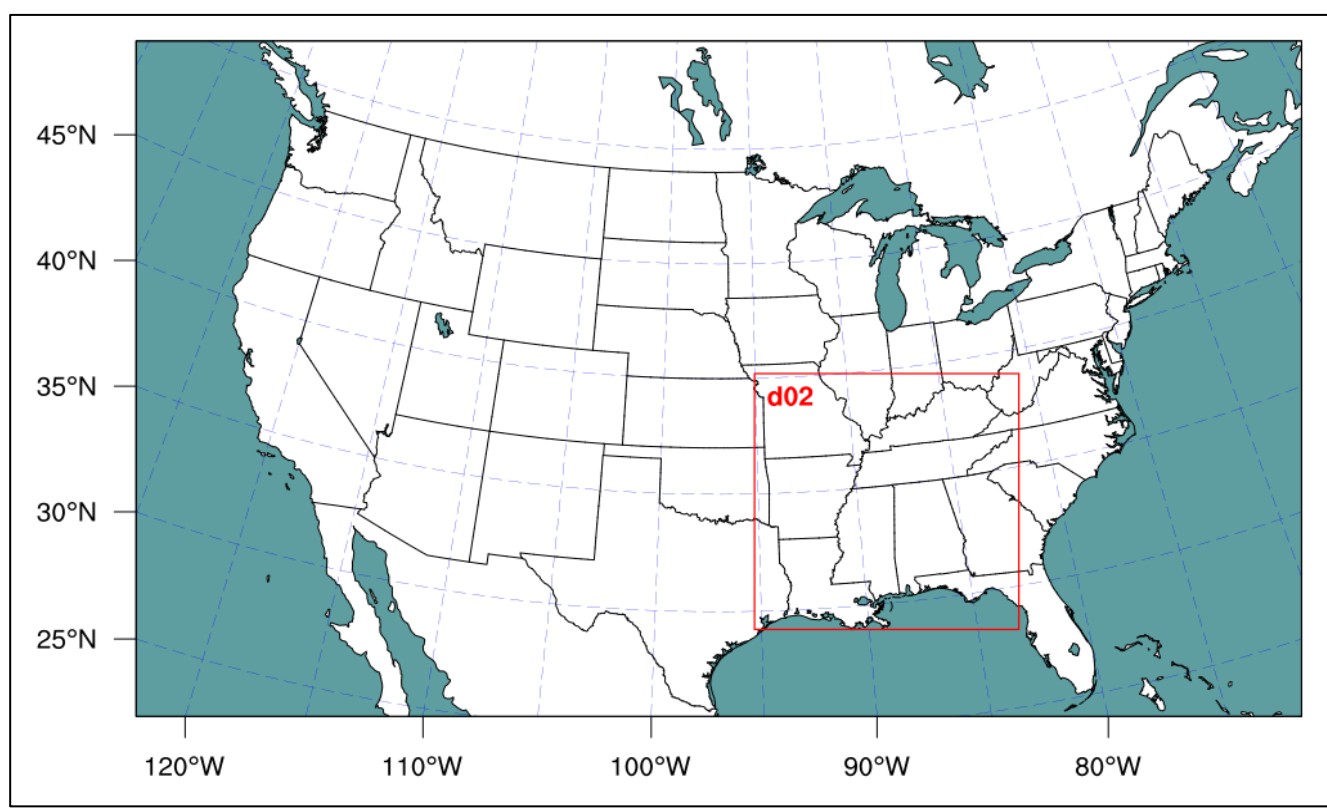

**Figure 4: The numerical domains of the NOMADSS simulations: the coarse domain has 442×265 grid points with 12 km grid cells, the nested domain, with 4 km grid cells, has 301×301 grid points.**






**Figure 5: Comparison between the 6-day (August 10th – 15th, 2015) average geopotential height (m) at 850 hPa obtained with (a) NCAR/NCEP reanalysis and (b) the WRF-Chem model. Comparison between the 6-day (August 10th – 15th, 2015) mean temperature at 995 hPa obtained from (c) NCAR/NCEP reanalysis, and (d) the WRF-Chem model.**




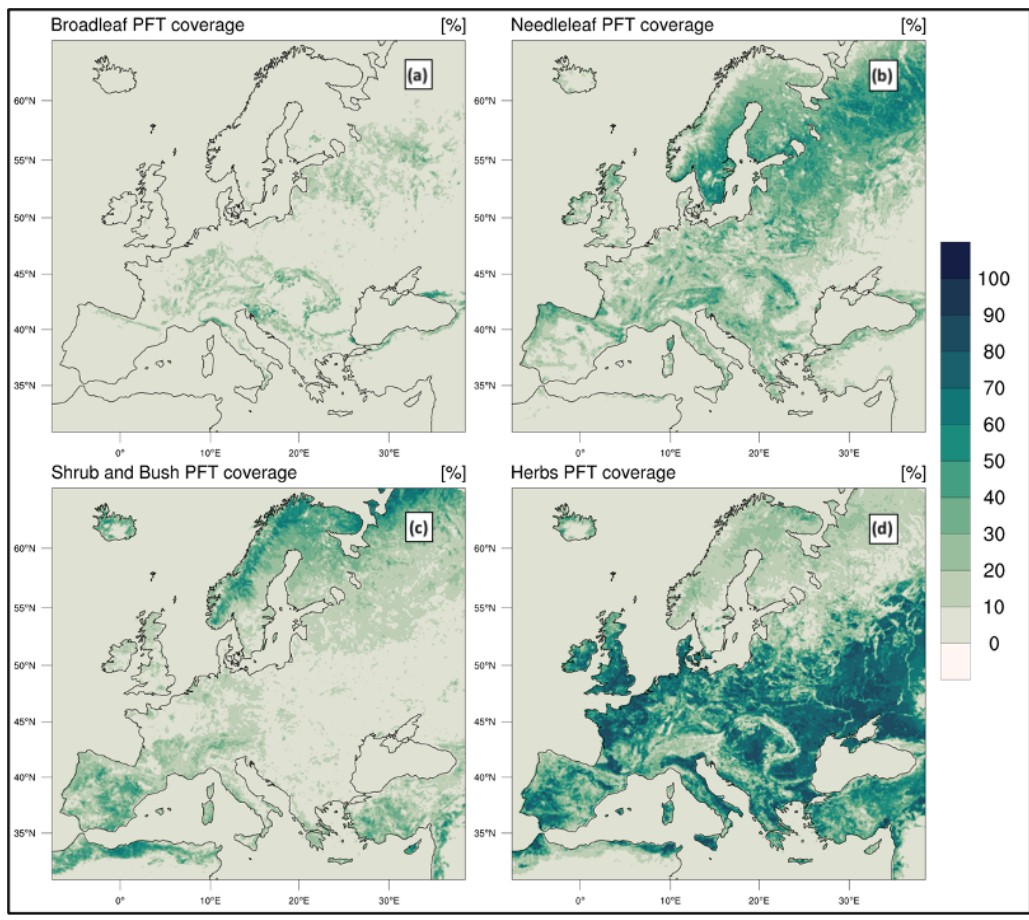

**Figure 6: Percentage coverage (%) of plant functional types (PFTs) classification included into MEGAN database, computed in August 2015. From the upper left map: (a) coverage of broadleaf trees (PFTP_HB), (b) needleleaf trees (PFTP_NB), broadleaf shrubs (PFTP_SB), and (d) grass and other (PFTP_HB).**






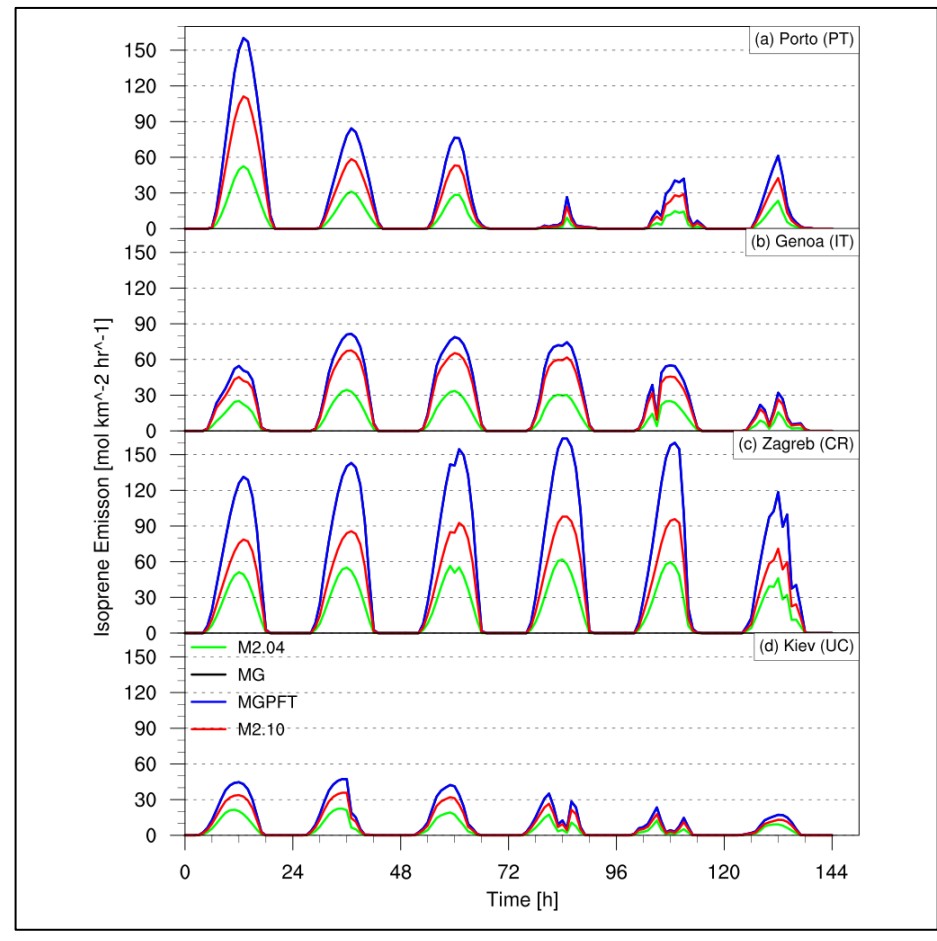

**Figure 7: Time series of isoprene emissions (mol km⁻² hr⁻¹) for different MEGAN algorithm configurations evaluated in 4 cities in Europe, namely (a) Porto, Portugal; (b) Genoa, Italy; (c) Zagreb, Croatia; (d) Kiev, Ukraine. The time period considered is from August 10th, 2015 at 0000 UTC to August 16th, 2015 at 0000 UTC. The green lines represent the control simulation (M2.04), the black lines indicate the activity factors (γ) updates (MG), the blue lines are representative of the PFTs emission factors updates (MGPFT), and the red lines show the isoprene emission factor as the emission factor of all the other compound classes (M2.10).**






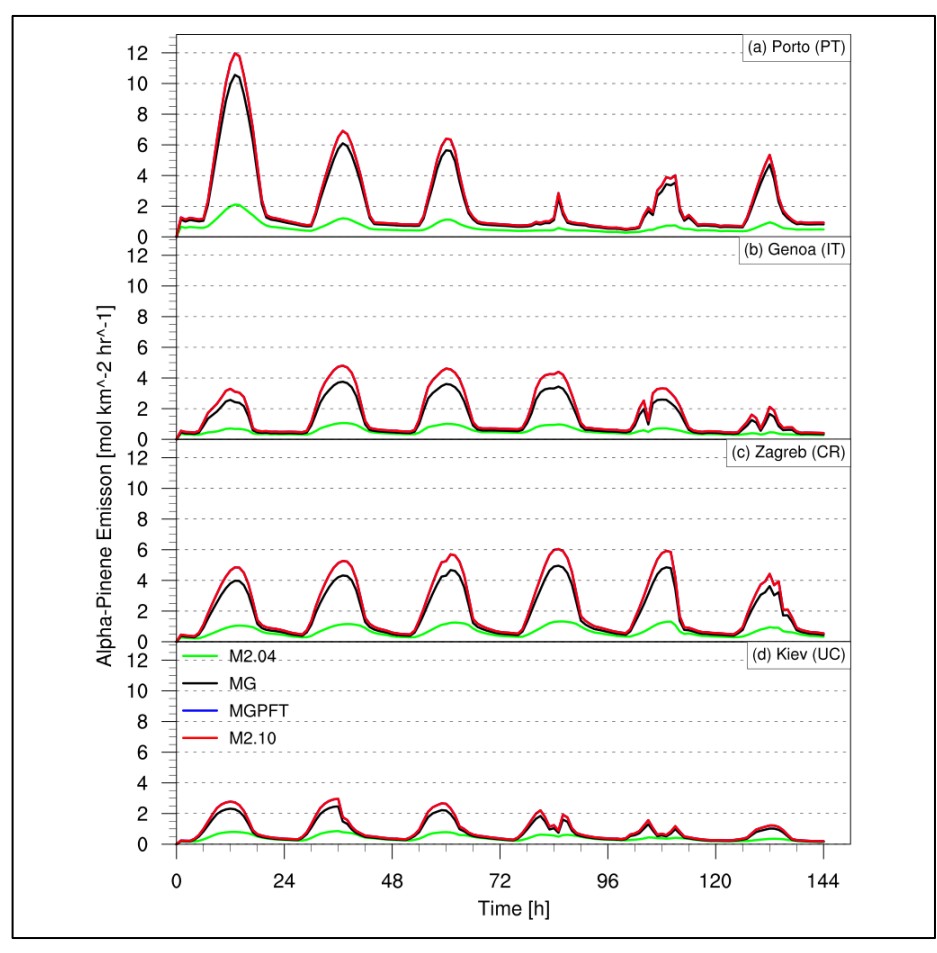

**Figure 8: Time series of α-pinene emissions (mol km$^{-2}$ hr$^{-1}$) for different MEGAN algorithm configurations evaluated in 4 cities in Europe, namely (a) Porto, Portugal; (b) Genoa, Italy; (c) Zagreb, Croatia; (d) Kiev, Ukraine. The time period considered is from August 10$^{th}$, 2015 at 0000 UTC to August 16$^{th}$, 2015 at 0000 UTC. The green lines represent the control simulation (M2.04), the black lines indicate the activity factors ($\gamma_i$) updates (MG), the blue lines are representative of the PFTs emission factors updates (MGPFT), and the red lines show the isoprene emission factor as the emission factor of all the other compound classes (M2.10).**



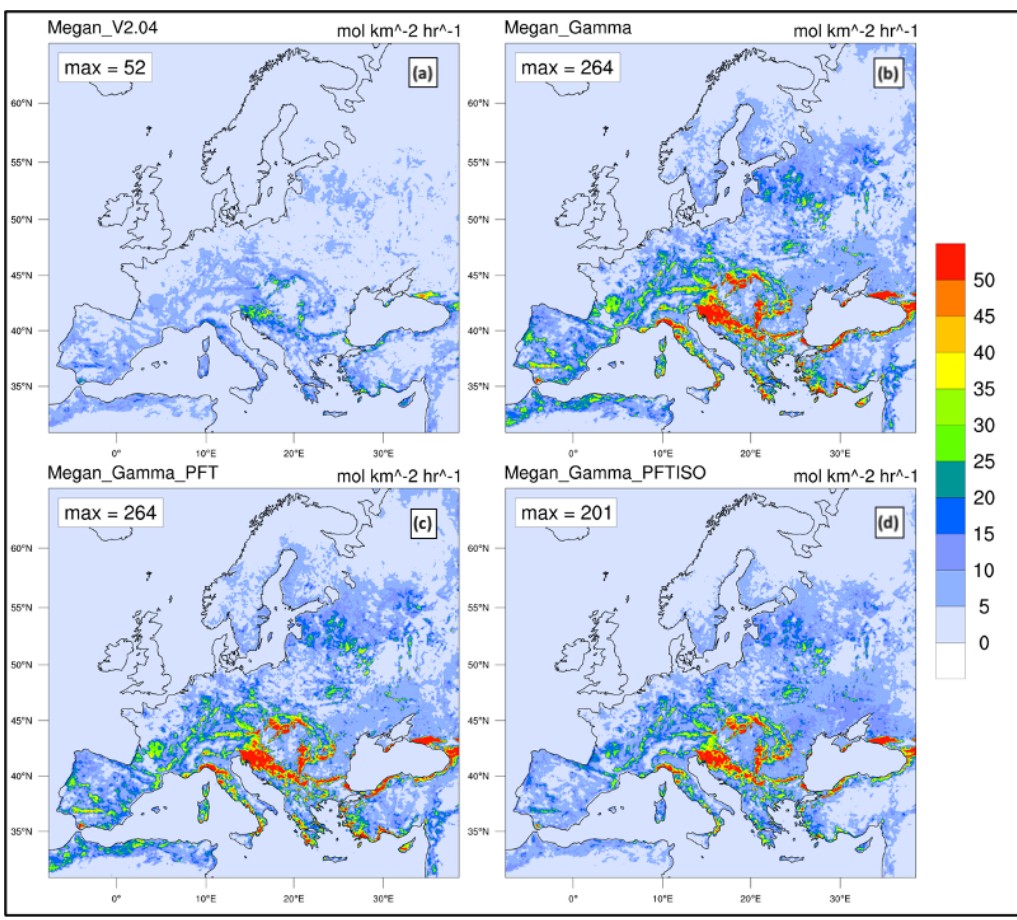


**Figure 9: The spatial distribution of isoprene emission (mol km$^{-2}$ hr$^{-1}$) calculated as average in the time period from August 10, 2015 at 0000 UTC to August 16, 2015 at 0000 UTC for the different MEGAN configurations, namely (a) control simulation (M2.04), (b) activity factors ($\gamma_i$) updated (MG), (c) PFTs emission factors updated (MGPFT), and (d) the isoprene emission factor updated (M2.10).**



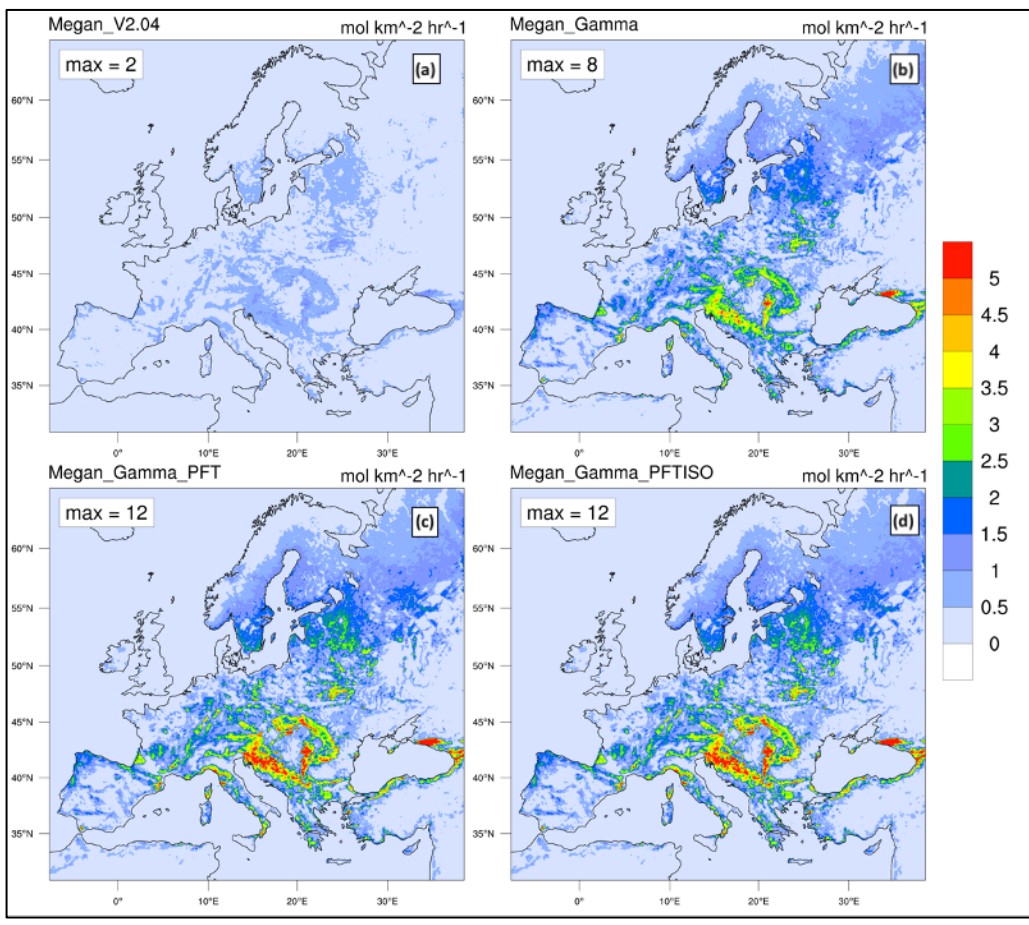


**Figure 10: The spatial distribution of α-pinene emission (mol km$^{-2}$ hr$^{-1}$) calculated as weekly average (from August 10, 2015 at 0000 UTC to August 16, 2015 at 0000 UTC) for the different MEGAN configurations, namely (a) control simulation (M2.04), (b) activity factors (γ$_i$) updated (MG), (c) PFTs emission factors updated (MGPFT), and (d) the isoprene emission factor updated (M2.10).**





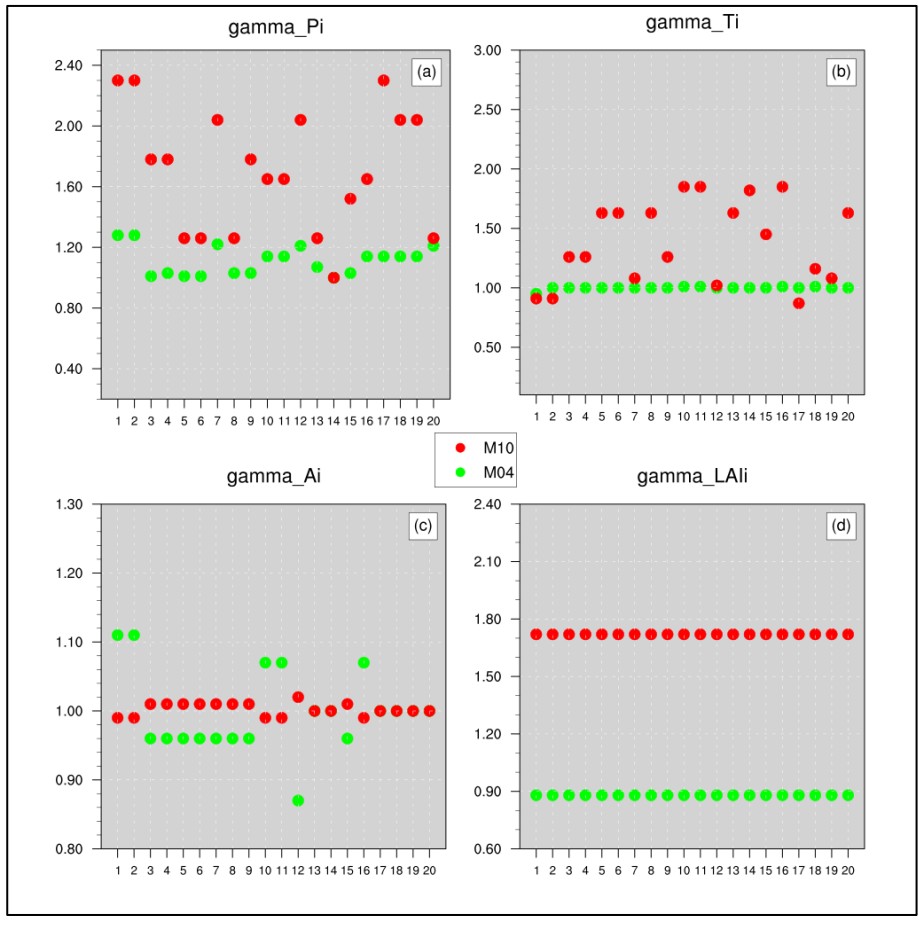

**1)**    **Figure 11: Comparison between M2.04 and M2.10 run of the emission activity factors (dimensionless) (a) photosynthetic photon flux density ($\gamma_P$, GAMMA_P), (b) temperature ($\gamma_T$, GAMMA_T), (c) leaf age ($\gamma_{age}$, GAMMA_A), and (d) leaf area index ($\gamma_{LAI}$, GAMMA_LAI) classified by classes compound (1. Isoprene, 2. Myrcene, 3. Sabinene, 4. Limonene, 5. 3-Carene, 6. t-β-Ocimene, 7. β-Pinene, 8. α-Pinene, 9. Other Monoterpenes, 10. α-Farnesene, 11. β-Caryophyllene, 12. Other Sesquiterpenes, 13. 232-MBO, 14. Methanol, 15. Acetone, 16. Carbon Monoxide, 17. Nitric Oxide, 18. Bidirectional VOC,**
885                        **19. Stress VOC and 20. other VOC). The factors refer to the city of Genoa (Italy) on August 13th (12:00 UTC), 2015.**



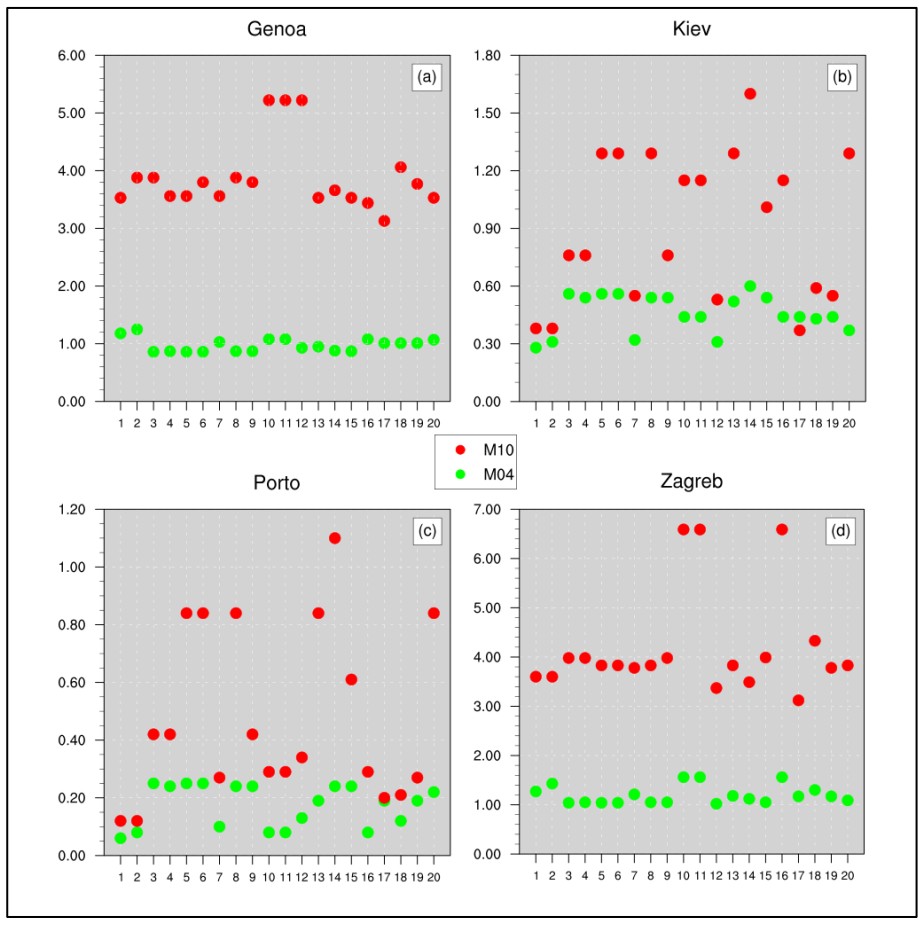

**Figure 12: Comparison between M2.04 and M2.10 run of the emission activity factors (dimensionless) for the city of (a) Genoa (Italy), (b) Kiev (Ukraine), (c) Porto (Portugal), and (d) Zagreb (Croatia), classified by classes compound (1. Isoprene, 2. Myrcene, 3. Sabinene, 4. Limonene, 5. 3-Carene, 6. t-β-Ocimene, 7. β-Pinene, 8. α-Pinene, 9. Other Monoterpenes, 10. α-Farnesene, 11. β-Caryophyllene, 12. Other Sesquiterpenes, 13. 232-MBO, 14. Methanol, 15. Acetone, 16. Carbon Monoxide, 17. Nitric Oxide, 18. Bidirectional VOC, 19. Stress VOC and 20. other VOC), on August 13th (12:00 UTC), 2015.**





905

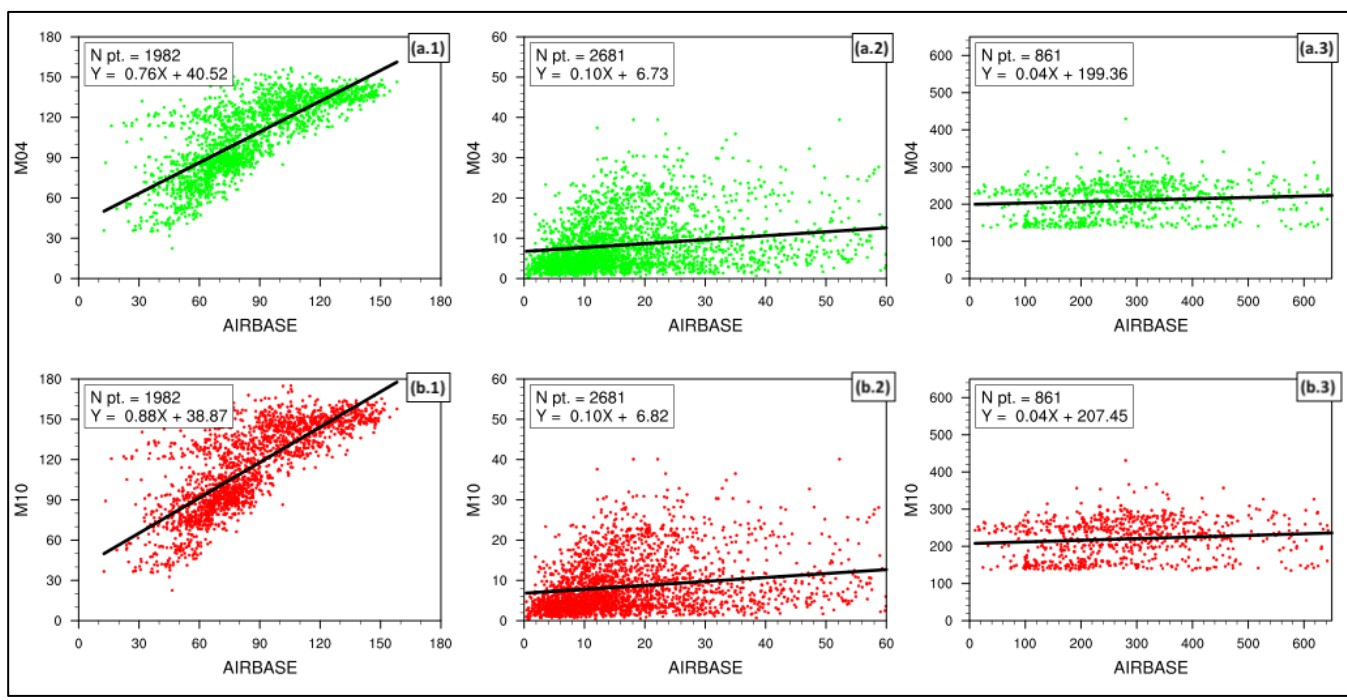

**Figure 13: Scatter plot and linear regression for the simulations M2.04 (a-green dots) and M2.10 (b-red dots) against the observed**
910 **(AirBase dataset) concentrations ($\mu$g m$^{-3}$) of (1) ozone, (2) nitrogen dioxide and (3) carbon monoxide. Each plot also shows the number of points recorded (N pt.) and equation for the line of best fit (Y). The values shown are the 8-hour daytime average, weekly mean from August 10$^{th}$, 2015 at 0000 UTC to August 16$^{th}$, 2015 at 0000 UTC.**



**Figure 14: Comparison between the (1) AIRBASE dataset of the mean day time (7 am – 6 pm UTC), (2) control simulation (M2.04 run) and (3) M2.10 run with all the MEGAN code updates concentrations of (A) O$_3$, (B) CO, and (C) NO$_2$ over the period from August 10$^{th}$ 00:00 UTC to 16$^{th}$ 00:00 UTC, 2015.**

Figure 15: The flight altitude (a - km), the temperature (b - K), the concentration of isoprene (c - ppb), methacrolein (MACR) (d - ppb), methyl vinyl ketone (MVK) (e – ppb), and ozone (f - ppb), for the second NOMADSS flight (rf02). The black line shows the C-130 aircraft measurements, the green and red lines indicate the WRF-Chem model results using MEGAN version 2.04 (M2.04 run) and MEGAN updated to the version 2.10 (M2.10 run), respectively. In panel b) the green line is not showed since it is overlapped by the red line, they have identical values.