# Peer review of "Comparison and evaluation of updates to WRF-Chem (v3.9) biogenic emissions using MEGAN"

_Geoscientific Model Development, 2022_

## Referee Comment (RC1)

**General Comments**

This paper describes the differences in BVOCs simulations using updated MEGAN. The concentrations of ozone, $NO_2$ and other trace gases over the two cases are also examined. Evaluation of the three updates made to the MEGAN coupled in WRF-Chem (v3.9) is useful to the model community. However, the value of this study has not been proved in the case that previous studies have already evaluated and compared the difference of V2.1 and V2.0 in WRF-Chem (Zhao et al., 2016; Zhang et al., 2021). It is not surprising that the updates to emission factors and activity factors would lead to the changes in BVOCs emissions. Therefore, it is important to quantitatively evaluate these differences and explain the reason. I appreciate that you collected the station and flight observations for evaluation. However, some analysis of OH, formaldehyde, and other necessary species may help explain the difference of the simulations. Most importantly, if the analysis found that the older version performances better against the observations than the newer version, the authors need to give some explanation and provide suggestion and guidance for further development or optimization of the model.

**Specific Comments**

1. Line 265-266: Please explain the difference of MG and MGPFT simulations in more details. Why is the activity factor related to the PFT emission factors?

2. Line 310-325: As the authors mentioned, BOVCs predicted by MEGAN are highly related to the environmental conditions. Therefore, there still need some comparisons of radiation, soil moisture, and other meteorological fields between the simulations and the observations (or reanalysis data) to confirm the reliability of the model.

3. Line 345-349: Adding similar plots of Figure 9 to show the PFT weighted emission factor (consider the PFT emission factor and PFT percentage) would make it clearer that the PFT percentage in four cities contributes to the difference of overall emission factors.

4. Line 392-393: Please explain a little more about the method of disaggregating station types here other than just citing a reference.

5. Line 405-406: Is soil $NO_x$ changed in different MEGAN simulations or do you turn off the soil $NO_x$ in these simulations? If soil NOx is different, the impact may not be from BVOCs only. In addition, please compare your results (the resulted impacts to $NO_2$ and CO are small) to other similar studies and provide some discussion.

**Technical corrections**

1. Line 31: It is not clear here about M2.04 and M2.10.

2. Line 195: There may be some errors in reference insertion.

3. Figure 11: "M10" and "M04" may cause confusion, please change to "M2.04" and

"M2.10" or give some explanation.

**References**

Zhao, C., M. Huang, J. D. Fast, L. K. Berg, Y. Qian, A. Guenther, D. Gu, M. Shrivastava, Y. Liu, S. Walters, G. Pfister, J. Jin, J. E. Shilling, C. Warneke (2016): Sensitivity of biogenic volatile organic compounds (BVOCs) to land surface parameterizations and vegetation distributions in California, Geosci. Model Dev., 9, 1959–1976, doi:10.5194/gmd-9-1959-2016, 2016.

Zhang, M.-S., C. Zhao, Yuhan Yang, Qiuyan Du, Yonglin Shen, Shengfu Lin, Dasa Gu, Wenjing Su, Cheng Liu: Modeling sensitivities of BVOCs to different versions of MEGAN emission schemes in WRF-Chem (v3.6) and its impacts over eastern China, Geosci. Model Dev., 14, 6155–6175, 2021.

---

## Author Comment (AC1)

**Comment 1**

*General Comments*

*This paper describes the differences in BVOCs simulations using updated MEGAN. The concentrations of ozone, $NO_2$ and other trace gases over the two cases are also examined. Evaluation of the three updates made to the MEGAN coupled in WRF-Chem (v3.9) is useful to the model community. However, the value of this study has not been proved in the case that previous studies have already evaluated and compared the difference of V2.1 and V2.0 in WRF-Chem (Zhao et al., 2016; Zhang et al., 2021). It isnot surprising that the updates to emission factors and activity factors would lead to the changes in BVOCs emissions. Therefore, it is important to quantitatively evaluate these differences and explain the reason. I appreciate that you collected the station and flight observations for evaluation. However, some analysis of OH, formaldehyde, and other necessary species may help explain the difference of the simulations. Most importantly, if the analysis found that the older version performances better against the observationsthan the newer version, the authors need to give some explanation and provide suggestion and guidance for further development or optimization of the model.*

*Answer to the general comments*

We thank the reviewer for the comprehensive comments, they are very helpful for improving the quality of the manuscript. One of the main objectives of the present paper is to help the community recognize the differences between different MEGAN model versions (i.e., 2.04 and 2.10) implemented within WRF-Chem and to provide guidance for next steps in developing MEGAN in chemistry transport models. We are glad that we got the message out.

We believe our paper is valuable because Zhao et al. (2016) investigated the sensitivity of WRF-Chem simulated BVOC emissions with different land surface schemes (i.e., Community Land Model version and Noah land surface model) considering also different vegetation maps. On the other hand, Zhang et al. (2021), used three versions of MEGAN (i.e., MEGAN v1.0, MEGAN v2.0, MEGAN v3.0). MEGAN v1.0 is the first model version coupled in WRF-Chem as it considers only the response of emission to radiation and temperature and MEGAN v2.0 is the second model version coupled with WRF-Chem that considers the emission factor of BVOCs for each grid calculated with a prescribed vegetation distribution and emission factor for each PFT (Guenther et al., 2006). Lastly, MEGAN v3.0 employed in the Zhang et al. (2021) study is updated from MEGAN v2.1 as implemented by Zhao et al. (2016). The main update in MEGAN v3.0 from MEGAN v2.1 is to consider the drought activity factor as an environmental forcing for biogenic emissions. Although Zhao et al. (2016) and Zhang et al. (2021) have

implemented MEGAN v2.1 in WRF-Chem with the CLM4 land model, (i.e., CLM surface scheme and associated subroutines in the physics and chemistry packages have been modified to be consistent with the MEGANv2.1 biogenic emission) in this work, we explore the effect of making simple changes to the existing WRF-Chem MEGAN v2.04 emissions scheme (Guenther et al., 2006). That is, the version implemented in WRF-Chem using the Noah land surface model, which is the same version called MEGAN v2.0 in the Zhang et al. (2021) paper). The version we present employs MEGAN emissions updates that can be used independently of the land surface model chosen. The changes made are consistent with MEGAN version described in Guenther et al. (2012).

We agree that it is not surprising that there are differences in modeling results among different versions of parameterization. We also expected that the update in the newer version can improve the model performance, which, however, is not true as we learned in this study. In the revised paper, we discuss and clarify the differences between MEGAN versions, some comparison with OH or formaldehyde, and recommend clearly that the MEGAN developers should re-examine the new coefficients. However, we disagree on providing improvements to MEGAN ourselves, as we do not have the expertise that the experimentalists have to guide possible improvements.

_Specific Comments_

**Line 265-266: Please explain the difference of MG and MGPFT simulations in more details. Why is the activity factor related to the PFT emission factors?**

Thanks for the clarification, the sentence "The third simulation (MGPFT) adds the changes in the activity factors due to the variation of the PFTs emission factors." is just meaningless, so we corrected to: "The third simulation (MGPFT) adds to the changes in the activity factors the variation of the PFTs emission factors."

**Line 310-325: As the authors mentioned, BOVCs predicted by MEGAN are highly related to the environmental conditions. Therefore, there still need some comparisons of radiation, soil moisture, and other meteorological fields between the simulations and the observations (or reanalysis data) to confirm the reliability of the model.**

Thanks for the suggestion. For the Europe case, we did analyze the temperature and geopotential height across Europe comparing WRF-Chem results with NCEP/NCAR reanalysis.
For final version of the paper, we plan to add new plots for solar radiation and precipitation.

*Line 345-349: Adding similar plots of Figure 9 to show the PFT weighted emission factor (consider the PFT emission factor and PFT percentage) would make it clearer that the PFT percentage in four cities contributes to the difference of overall emission factors.*

Thank for the very valuable comment. You gave us a very delightful idea: we replaced the figure 6 (i.e., the plots with the % of PFTs) with the PFT emission weighted factor plots, for Isoprene and Alpha-pinene (Figure 8 and Figure 10).

[Figure]

**Figure 8: PFT weighted emission factor (PFT emission factor and PFT percentage) (ug km$^{-2}$ hr$^{-1}$) of isoprene, computed in August 2015. The emission factor values used are from the** Error! Reference source not found. **(2.10 column). From the upper left map: (a) PFT weighted emission factor of broadleaf trees (PFTP_BT), (b) needleleaf trees (PFTP_NB), broadleaf shrubs (PFTP_SB), and (d) grass and other (PFTP_HB).**

[Figure]

**Figure 10: PFT weighted emission factor (PFT emission factor and PFT percentage) (ug km⁻² hr⁻¹) of α-pinene, computed in August 2015. The emission factor values used are from the** Error! Reference source not found. **(2.10 column). From the upper left map: (a) PFT weighted emission factor of broadleaf trees (PFTP_BT), (b) needleleaf trees (PFTP_NB), broadleaf shrubs (PFTP_SB), and (d) grass and other (PFTP_HB).**

*Line 392-393: Please explain a little more about the method of disaggregating station types here other than just citing a reference.*

We introduced the meaning of the station type: urban, suburban and rural surface station, now the sentence is:

Since discrepancies between modelled and measured values might be related to the type and location of a measurement station, the selected stations were also disaggregated into categories based on the study done by Henne et al., 2010, which includes a more complete analysis of the surroundings of each station. The alternative classification (see Supplement S3) provides three class station types: urban, suburban and rural surface stations. Urban means a continuously built-up urban area (buildings with at least two floors), the built-up area is not mixed with non-urbanized areas; suburban area is largely built-up urban area, it means, contiguous settlement of detached buildings of any size, the built-up area is mixed with non-urbanized areas (e.g., agricultural, lakes, woods). All areas, that do not achieve the criteria for urban or suburban areas, are defined as rural areas.

*Line 405-406: Is soil NO$_X$ changed in different MEGAN simulations or do you turn off the soil NO$_X$ in these simulations? If soil NOx is different, the impact may not be from BVOCs only. In addition, please compare your results (the resulted impacts to NO2 and CO are small) to other similar studies and provide some discussion.*

Thanks for the question, it is a very interesting observation.
The soil NOx does not change in different MEGAN simulations, since the value of PFT percentage (i.e., PFTP_BT, PFTP_NB, PFTP_SB and PFTP_HB) from the wrfbiochemi file remains unchanged.
Following your suggestions, we added to the paper the following comments:

For the different model runs anthropogenic, biogenic and biomass burning NOx emissions did not vary. Specifically, soil NOx emissions were evaluated with MEGAN as a function of environmental variables (i.e., temperature and vegetation types) that were the same for each model run. Therefore, no substantial changes were noted for the NOx concentration levels for the different model runs. Recent studies regarding the effects of NOx soil emissions on O3 levels in California (USA) (Sha et al., 2021) and Europe (Visser et al., 2019) have pointed out that NOx levels were underestimated with large biases because of the low NOx soil emissions estimated with WRF-Chem/MEGAN. NOx soil emissions are important both on the tropospheric NOx budget and surface O3 level perspectives (Sha et al., 2021). Considering that the model runs with increases in BVOC emissions showed higher O3 levels, it is likely that the O3 formation was not NOx limited.

The increase in CO concentration values is small compared to the increase observed for isoprene, because both emission factor and emission activity factor of isoprene are higher in 2.10 version compared to 2.04 version.
We added to the paper the following comments:

MEGAN estimates carbon monoxide emissions as biogenic emission class unlike NO$_x$ soil emissions. Higher CO emissions were noted for the MG simulation compared to the control run (M2.04) because of the changes in emission activity factors ($\gamma_i$). As reported in **Error! Reference source not found.**, CO emission factor differs between MG and MGPFT runs, with a lower value for MGPFT (600 CO μg m$^{-2}$ hr$^{-1}$) compared to MG (1000 CO μg m$^{-2}$ hr$^{-1}$). Moreover, the higher emission activity factor and lower CO emission factor in MGPFT compared to the control run resulted in only slight differences in CO levels between the two runs. This results in the different model runs showing slight variations in CO levels.

Technical corrections

**Line 31: It is not clear here about M2.04 and M2.10.**

I changed the sentence as follow, I hope it is clearer:

The comparison between the modeled data and aircraft observations shows that isoprene mixing ratios measured agree well with M2.04 simulation but are overpredicted considerably by the M2.10 simulation.

**Line 195: There may be some errors in reference insertion.**

Yes, it is a reference error, I corrected it.

**Figure 11: "M10" and "M04" may cause confusion, please change to "M2.04" and "M2.10" or give some explanation.**

I modified the caption of figure 11, 12 and 13:

Figure 1: Comparison between M2.04 (M04) and M2.10 (M10) ….

Figure 2: Comparison between M2.04 (M04) and M2.10 (M10) ….

Figure 3: Scatter plot and linear regression for the simulations M2.04 (M04 - a-green dots) and M2.10 (M10 - b-red dots) ….

**References**

Guenther, A., Karl, T., Harley, P., Wiedinmyer, C., Palmer, P. I. and C., G.: Estimates of global terrestrial isoprene emissions using MEGAN, Atmos. Chem. Phys. Discuss., 6(1), 107–173, doi:10.5194/acpd-6-107-2006, 2006.

Guenther, A., Jiang, X., Heald, C. L., Sakulyanontvittaya, T., Duhl, T., Emmons, L. K. and Wang, X.: The model of emissions of gases and aerosols from nature version 2.1 (MEGAN2.1): An extended and updated framework for modeling biogenic emissions, Geosci. Model Dev., 5(6), 1471–1492, doi:10.5194/gmd-5-1471-2012, 2012.

Zhang, M., Zhao, C., Yang, Y., Du, Q., Shen, Y., Lin, S., Gu, D., Su, W. and Liu, C.: Modeling sensitivities of BVOCs to different versions of MEGAN emission schemes in WRF-Chem (v3.6) and its impacts over eastern China, Geosci. Model Dev., 14(10), 6155–6175, doi:10.5194/gmd-14-6155-2021, 2021.

Zhao, C., Huang, M., Fast, J. D., Berg, L. K., Qian, Y., Guenther, A., Gu, D., Shrivastava, M., Liu, Y., Walters, S., Pfister, G., Jin, J., Shilling, J. E. and Warneke, C.: Sensitivity of biogenic volatile organic compounds to land surface parameterizations and vegetation distributions in California, Geosci. Model Dev., 9(5), 1959–1976, doi:10.5194/gmd-9-1959-2016, 2016.

---

## Author Comment (AC2)

**Comment 2**

***General Comments***

*The authors made an effort to update and test the MEGAN biogenic model coupled to the WRF-Chem model. The major problem is that the results are slightly worse than the previous version and the question upon what should be done to improve the model remains unanswered. Also, this poses a question, why to use a new version instead of the older one? In addition, the US case lacks the proper statistical verification and should be done. However, the paper provides the information on the new version and can be valuable mid-step towards the further improvement of the model.*

*Except for this problem, the paper is well written and organized. However, I provide a list of minor comments below.*

***Answer to the general comments***

We thank the reviewers for their sincere and constructive comments; we are pleased that our effort is recognized.

We started the process by comparing the two versions of MEGAN, updating different components of its modules in WRF-Chem (i.e., emission activity factor and emission factor), with the purpose to recognize which parameters generate major differences in terms of BVOC emission and ozone concentration. We expected that the update in the newer version could improve the model performance, which, however, was not the case. In our paper, we do not want to recommend one version over another, but instead aim to highlight the shortcomings of BVOC emission modeling that future experimental and modeling studies can address.

The reviewer makes a good point that the US case lacks the proper statistical verification. In the final version, we will add statistical evaluation for the US case.

*__Minor comments__*

*__CA.1) Abstract is too long and includes too many details, particularly part after line 21. Try to point out the main results and make the abstract shorter and/or remove part afterline 21 in introductions.__*

We have revised the abstract to be more concise and to highlight the paper's main results and conclusions. The new abstract is the following.

Biogenic volatile organic compounds (BVOCs) emitted from the natural ecosystem are highly reactive and thus can impact air quality and aerosol radiative forcing. BVOC emission models (e.g., Model of Emissions of Gases and Aerosols from Nature, MEGAN) in global and regional chemical transport models still have large uncertainties in estimating biogenic trace gases, because of uncertainties in emission activity factors, specification of vegetation type, and plant emission factors. This study evaluates a set of updates made to MEGAN v2.04 in the Weather Research and Forecasting model coupled with chemistry (WRF-Chem version 3.9). Our study considers four simulations for each update made to MEGAN v2.04, (i) a control run with no changes to MEGAN; (ii) a simulation with the emission activity factors modified following MEGAN v2.10; (iii) a simulation considering the changes to the plant functional type emission factor; and (iv) a simulation with the isoprene emission factor calculated within the MEGAN module instead of prescribed by the input database. We evaluate two regions, Europe and the Southeast United States, by comparing WRF-Chem results to ground-based monitoring observations in Europe and aircraft observations obtained during the NOMADSS field campaign. We find the updates to MEGANv2.04 in WRF-Chem caused overpredictions in ground-based ozone concentrations in Europe and in isoprene mixing ratios compared to aircraft observations in the Southeast US. The update in emission factors caused the largest biases. These results suggest that further experimental and modeling studies should be conducted to address potential shortcomings in BVOC emission models.

*__CA.2) Line 24. Sentence The updated MEGAN model… is not clear, what does the estimated BVOC emissions refer to? Please rephrase the sentence__*

We deleted the sentence since the abstract is changed.

***CA.3) Line 27. Is the bias obtained from comparison of measurements and modeled values of ozone? The sentence is not clear, rephrase***

The bias is obtained from comparison of control run and simulations with MEGAN changes.

***C1.1) In Introduction, the 'discussion' regarding the different modeling approach is missing the information about the capability of the models to simulate the BVOC emissions and O3 concentrations. Can you find the information about it and explain in general what is the success of modeling BVOC emissions and O3 concentrations and what are the uncertainties (both in modeling and measurements). Add this part between lines 65 and 80.***

Following your comment, we added the following sentences:

Line 58:

Several gaps in BVOC emission modelling were addressed in recent releases of MEGANv3 (Guenther et al. 2017) and MEGANv3.1 (Guenther et al. 2019), including BVOC emissions (i) accounting for sub-grid vegetation distribution in addition to the dominant vegetation type; (ii) induced by environmental stresses (i.e. extreme weather and air pollution events).

Line 82:

In the work by Wang et al. (2021), the impact of BVOC emissions evaluated with MEGANv3.1 on O3 concentrations simulated with WRF/CAMx varied highly with the drought configurations, with the highest BVOC contribution to O3 concentrations for not including drought stress.

Despite of the studies of BVOC and O3 model intercomparisons or sensitivity to different schemes, different authors (Messina, et al. 2016; Zhang et al. 2021) pointed out the need for more measurement campaigns of BVOC emissions to validate BVOC model results.

***C1.2) Line 50. What are the other meteorological parameters used in the MEGAN model?***

I added the soil moisture to the other meteorological parameters already mentioned (temperature and solar radiation):

This model estimates the emissions considering meteorology (e.g., temperature, solar radiation, and soil moisture), leaf area index (LAI), and plant functional type (PFT) as driving variables, with higher emissions occurring for higher temperature, transmission of photosynthetic photon flux density, and LAI.

***C1.3) Lines 50-60 should be moved in the model description chapter***

Thanks for the great suggestion. We moved those lines as an introduction to section 2.

**2. Materials and Methods**

The MEGAN model estimates the emissions considering meteorology (e.g., temperature, and solar radiation), leaf area index (LAI), and plant functional type (PFT) as driving variables, with higher emissions occurring for higher temperature, transmission of photosynthetic photon flux density, and LAI. MEGAN v2.0 was used for analyzing the impact of biogenic emissions with potential future increases in ambient temperature on ozone levels (Im et al., 2011), aerosol levels and chemical compositions (Im et al., 2012). Building on MEGAN v2.0 (G06) and MEGAN v2.02 (Sakulyanontvittaya 55 et al., 2008), Guenther et al. (2012) (G12 hereafter) introduced additional compounds, emission types, and controlling processes with MEGAN v2.1. In MEGAN v2.1, the emission factors are adjusted to consider that the measured net flux of BVOC compounds above the vegetation canopy does not involve the dry deposition flux, so that the net primary emissions would be higher (e.g., up to a few percent for isoprene). To better depict the variability of isoprene emission within a PFT category, MEGAN v2.1 allows specific PFT emission factors for each vegetation type.

**2.1 Updates to MEGAN v2.04 in WRF-Chem**

***C1.4) Line 61-65. Sentence beginning with Zao et al. (2016)... is confusing, particularly after , "namely the Community Land Model,..", what is the relation between Noah, CLM4 and MEGAN? Explain and separate this sentence into 2 sentences***

Thanks for asking clarification, this is a crucial point, so we explained better, from this:

Zhao et al. (2016) investigated the sensitivity of WRF-Chem simulated BVOC emissions with different land surface schemes, namely the Community Land Model version 4.0 (CLM4 - Oleson et al., 2010; Lawrence et al., 2011) and the Noah land surface model (Niu et al., 2011), for MEGAN v2.0, and considering also different vegetation maps for MEGAN v2.1 implemented into the CLM4.

To this:

Zhao et al. (2016) used two versions (v2.04 and v2.1) of MEGAN in order to investigate the sensitivity of WRF-Chem simulated BVOC emissions with different land surface schemes: the Community Land Model version 4.0 (CLM4 - Oleson et al., 2010; Lawrence et al., 2011) and the Noah land surface model (Niu et al., 2011). The land surface schemes quantify land surface processes, their effect on near-surface meteorological conditions, and consequently the simulated BVOC emissions and concentrations. One major difference between the Noah land surface model and CLM4

is that they use different vegetation maps and this affects BVOC emissions.

**C1.5) Line 65. Which authors, there are several authors cited?**

It is referred to Zhao et al. (2016) authors.
The phrase has been replaced to now say, "Zhao et al. (2016) found …".

**C1.6) Line 67. What do you mean by "consistent variations in BVOC emissions predicted with MEGAN v2.1."? Paraphrase this sentence**

Thanks, the sentence is poorly phrased. I changed from:

These authors found that BVOC emissions modelled with MEGAN v2.0 were negligible between the two runs with different land surface schemes and one type of vegetation map, whereas considering the same land surface scheme with different vegetation maps induced consistent variations in BVOC emissions predicted with MEGAN v2.1.

To:

Zhao et al. (2016) found that BVOC emissions modelled with MEGAN v2.04 were negligible between the two runs with different land surface schemes and the same vegetation map, whereas considering the same land surface scheme with different vegetation maps leads to large differences in simulated BVOC emissions predicted with MEGAN v2.1.

**C1.8) Line 71. What do you mean by different meteorological drivers? Aren't the meteorological drivers the same for the BVOC emissions?**

Following your comment, we revised from:

These authors found values of isoprene and other compounds at the regional scale similar to the findings from previous studies (Sindelarova et al., 2014; Messina et al., 2016) that used different meteorological drivers, thus confirming that the emission factor and PFT distributions determine the spatial emission distribution in MEGAN.

To:

Henrot et al. (2017) found the emission factor and PFT distributions most strongly determine the spatial emission distribution in MEGAN in agreement with other previous studies that used different meteorological models (Sindelarova et al., 2014; Messina et al., 2016).

**C1.9) Line 78. What do you mean by five species?**

In Europe, the isoprene emission is dominated by three Quercus species, and the

monoterpene by five species. The sentence is not well structured, we changed from:

A few species dominate the total isoprene and monoterpene emissions in European forests, with three Quercus species and five species contributing to 66 % and 80 % of total isoprene and monoterpene emissions, respectively (Keenan et al., 2009).

To:

A few tree species dominate the total isoprene and monoterpene emissions in European forests, with three Quercus species and five types of tree species contributing to 66 % and 80 % of total isoprene and monoterpene emissions, respectively (Keenan et al., 2009).

**C1.10) Line 82. Can you be more specific about the evolution of MEGAN versions, why do you use version 2.04, while version 2.1 is the latest version in WRF 4.3.**

Thanks for suggestion, we changed the text, following your comment, from:

Although Zhao et al. (2016) implemented MEGAN v2.1 in WRF-Chem with the CLM4 land model, it did not become part of the community version of WRF-Chem until spring 2021 with the release of WRF version 4.3; the CLM surface scheme and associated subroutines in the physics and chemistry packages have been modified to be consistent with the MEGAN v2.1 biogenic emission. Here, we explore the effect of making simple changes to the existing WRF-Chem MEGAN v2.04 emissions scheme in WRF-Chem to provide MEGAN updates that can be used independently of land surface model chosen.

To:

Zhao et al. (2016) have implemented MEGAN v2.1 in WRF-Chem with the CLM4 land model; CLM surface scheme and associated subroutines in the physics and chemistry packages have been modified to be consistent with the MEGAN v2.1 biogenic emission. These changes become part of the community version of WRF-Chem in 2021 with the release of WRF version 4.3. In our work, which we performed before WRF version 4.3 was available, we use WRF-Chem version 3.9, to explore the effect of making changes to the existing WRF-Chem MEGAN v2.04 emissions scheme. Because we modified the MEGAN v2.04 code, our method results in having changes that can be used with the Noah land surface model.

*C1.11) Line 85. Put G12 instead Guenther et al. (2012). Also check this issue in other places in text for consistency*

Ok, thanks. Done.

*C1.12) Line 85. Again, issue with versions of Megan, it is not clear if you use version 2.1 or 2.04 in this study, later you suggest that the comparison was made between versions 2.0 and 2.1…*

To be even clearer, we updated MEGAN v2.04 with the equations of MEGAN v2.10.

*C1.13) Line 91. Add a link to the AirBase database. C1.14) Line 100. Add a link to the SAS.*

Ok, done.

*C2.1) As I understand, you made an update of version 2.04, and the updated version is 2.10. Add sentence in the beginning of chapter 2.1.*

See comment C2.9).

*C2.2) Line 114. To follow the explanation of the emission activity factors, set them in order as in eq. 1, and use brackets i.e., ð ¾LAI (leaf area index) ...etc. Apply this approach further in text if needed.*

I changed the order of the equation to the same of line 114, to follow the explanation, as you suggested, thanks.

Unfortunately, probably due to the error in the request, we couldn't understand where you would like we use the brackets.

*C2.3) Line 117. No need for typing plant functional type (PFT), use only PFT if abbreviation is already introduced in text before*

Ok, thanks for suggestion.

*C2.4) Line 117-118. Explain how the deviation from standard conditions is taken into account.*

Thanks for clarification, I added some standard conditions details. I modified the sentence from:

The emission rate (EM) is calculated for each plant functional type (PFT), added up to estimate the total emission at each model grid cell, and corrected taking into account the deviation from the standard condition ($\gamma$ and $\rho$ parameters).

To:

The emission rate (EM) is calculated for each PFT, added up to estimate the total emission at each model grid cell, and corrected taking into account the deviation from the standard condition ($\gamma$ and $\rho$ parameters). The factor $\gamma$ and $\rho$ are equal to unity at standard conditions (e.g., air temperature 303 K, humidity 14 g kg$^{-1}$, wind speed 3 m s$^{-1}$, and soil moisture 0.3 m$^3$ m$^{-3}$), while they are different from unity at other temperatures, humidity, wind speed, and soil moisture.

*C2.5) Line 128. Replace influencing with decreasing.*

Ok, done.

*C2.6) Line 129. What is the meaning of the sentence "The integration of MEGAN in CTMs (e.g., temperature, solar radiation, and soil moisture)."? Rephrase*

The meaning is intended to be, in the CTM models the interpolation of meteorological and MEGAN parameters, allows examinations between BVOC emissions and the surrounding environment. The sentence is not clear at all, I rephrased this:

The integration of MEGAN in CTMs (e.g., temperature, solar radiation, and soil moisture) allows examinations of interactions between BVOC emissions, the surrounding environment, and the canopy itself.

To:

The integration of MEGAN with CTMs parameters (e.g., temperature, solar radiation, and soil moisture) allows an improved analysis of interactions between BVOC emissions, the surrounding environment, and the canopy itself.

**C2.7) Line 133. What is the meaning of the local state "climate"? Rephrase in more precise manner.**

The local state is intended as the environmental conditions dependent on location and season, while the climate as the environmental effects caused by larger scale effects. I rephrased from:

Overall, the BVOC emissions is a product of both the local state (temperature and PPFD) and the "climate" (soil moisture and heat waves/drought), hence the emissions are a function of both the instantaneous temperature and the temperature averaged over 1–10 days.

To:

Overall, the BVOC emissions is a product of both the local weather at the time of simulation (i.e., temperature, humidity, and PPFD), and on long-term conditions, such as the conditions over the past month (i.e., based on seasonal conditions like soil moisture and heat waves or drought), hence the emissions are a function of both the instantaneous temperature and the temperature averaged over 1–10 days.

**C2.8) Line 139. What is the meaning of the sentence after "included it in the light,.."? Rephrase.**

Guenther et al. (2006) developed a parameterized canopy environment emission activity (PCEEA) algorithm to reduce computational cost. The PCEEA algorithm includes equations for the light, temperature and environment response emission activity ($\gamma_{LAI}$, $\gamma_P$ and $\gamma_T$). We changed the sentence from:

To minimize computational costs, G06 developed a parameterized canopy environment emission activity (PCEEA) algorithm as an alternative to calculating all variables at each canopy depth and included it in the light, temperature and canopy environment response emission activity factors in MEGAN v2.04.

To:

To minimize computational costs, G06 developed a parameterized canopy environment emission activity (PCEEA) algorithm as an alternative to calculating all variables at each canopy. The PCEEA procedure includes algorithms for the solar radiation, temperature and canopy environment response emission activity factors in MEGAN v2.04.

**C2.9) Line 140. So, you used version 2.04 and updated it to version 2.10? Rephrase and move to section 2.1**

Thanks for suggestion, I moved the lines in the section 2.1, it is clearer now.

**C2.10) Line 146. Delete "are comprised".**

Ok, done.

**C2.11) Line 147. Response emission activity should be mentioned after eq 1., line 115 … here use symbol only. Also, refer to the Tab. 1 since the equation with sine can be found there.**

Ok, done.

**C2.12) Line 148. Class compound or compound class? Also check line 169.**

Compound class, we checked and corrected all document, thanks.

**C2.13) Eq 2, index "i" stands for each compound class? Specify.**

We have specified before the equation 2:

For each (i-th) compound class, the updated emission activity factor accounting for the PPFD variations is changed to the following equations:

**C2.14) Line 150. Correct the part "...the Ps the…".**

Ok, done.

**C2.15) Line 152. The sentence "This new version…" is confusing. Instead of "new version" and "updated version" use the v2.04, or v2.1. Which version uses swdown and which mwdown?**

The swdown and mwdown are both used from the new version, v2.10.  The v2.04 used the sin of solar angle to calculate the temperature emission factors. We replaced the sentence:

This new version code calculates the γp with the photosynthetic photon flux density

using the internal variable "swdown": the downward solar radiation (W m-2). P24 and P240 are the average PPFD of the past day and the past ten days, nevertheless, in the modified code, they are both equal to "mswdown" variable: the downward solar radiation (W m-2) of previous month (G12).

To:

The version 2.10 calculates the $\gamma_p$ with the photosynthetic photon flux density using the internal variable "swdown": the downward solar radiation (W m$^{-2}$). $P_{24}$ and $P_{240}$ are the average PPFD of the past day and the past ten days, nevertheless they are both equal to "mswdown" variable: the downward solar radiation (W m$^{-2}$) of previous month (G12).

**C2.16) Line 158. Are all equations from Guenther et al. (2006), also use G06 instead of citation, also in line 171.**

Ok, done.

**C2.17) Eq 9. Use 0.05 instead of 0,05**

Ok, done.

**C2.18) Line 164. How is the T determined/calculated in the model? Add explanation**

We added the following explanation:

T is the leaf temperature (K) taken as the air temperature at 2 m (=T2) calculated by WRF at the grid point;

**C2.19) Line 175. Perhaps write "..values of T24 and T240 are estimated.." instead of "..value of T24 and T240 is estimated.."?**

Ok, done.

**C2.20) Lines 179-180. How is this in agreement with the statement in line 126: "with young leaves emitting no isoprene and mature leaves emitting isoprene maximally"?**

I corrected the line 137 from:

A leaf's capacity to emit isoprene is also influenced by leaf phenology, with young leaves emitting no isoprene and mature leaves emitting isoprene maximally.

To:

A leaf's capacity to emit isoprene is also influenced by leaf phenology, with very young leaves emitting no isoprene and mature leaves emitting isoprene maximally.

And the lines 189-190 from:

The canopy isoprene-emitting capability is also influenced by the leaf age. An increase in foliage is assumed to imply a higher production of isoprene (young leaves), whereas decreasing foliage is associated to less production of isoprene (old leaves).

To:

The canopy isoprene-emitting capability is also influenced by the leaf age. An increase in foliage is assumed to imply a growing production of isoprene (young leaves), whereas decreasing foliage is associated a production reducing of isoprene (old leaves).

**C2.21) Line 195, (Error! Reference source not found. Table 1)? Also replace citation with G12.**

Ok, done.

**C2.22) Equations below line 203, eq. numbers 13,14,... instead 1,2,3? Change.**

Ok, done.

**C2.23) Line 211. Explain $C_{ce}$ and canopy environmental model.**

Ok, we modified from:

$C_{ce}$ (=0.57) is a value dependent on the canopy environment model.

To:

Cce is a value dependent on the canopy environment model being used. WRF-AQ (Weather Research Forecast – Air Quality) canopy environment model uses a value of 0.57 (G12).

**C2.24) line 221, use G12...Table 2 is from G12 or from this paper, or both?**

We changed the sentence, following you comment, from:

Table 2 shows the new emission factors (µg m-2 hr-1) applied, for each type of plants with comparisons to the old values.

To:

The updated emission factors for the four PFTs, and their previous value from MEGAN v2.04, are shown in Table 2.

**C3.1) Figure 3 has some issues; the colorbar has no units, the x value is not explained in the caption and has no units. Also, it is unclear how the maps of the isoprene emission factors are obtained in regard to the previously mentioned options (prescribed and eq 1), explain...**

It is not very easy to read. The unit of color bar is (mol km-2 h-1), and the isoprene emission factors are from the control simulation (i.e., with MEGAN v2.04). The x values represent the isoprene mixing ratios, (part per trillion by volume) along the flight tracks. We modified the caption from:

Figure 1: Flight tracks and the relative aircraft-based measurements of isoprene concentrations (pptv) under the Southern Oxidant and Aerosol Study plotted over the different maps of isoprene emission factors (mol km-2 h-1) from WRF-Chem output, for each day of the research flight at 3:00 pm local time (20:00 UTC), namely (a) rf01: 03/6/13; (b) rf02: 05/6/13; (c) rf03: 08/6/13; (d) rf04: 12/6/13; (e) rf05: 14/6/13.

To:

Figure 2: The x values (i.e., colored dots) denote the isoprene mixing ratios (pptv) along the aircraft flight tracks plotted over the different maps of isoprene emission factors (mol km-2 h-1) from M2.04 simulation. Results are for each research flight day at 3:00 pm local time (20:00 UTC), namely (a) rf01: 03/6/13; (b) rf02: 05/6/13; (c) rf03: 08/6/13; (d) rf04: 12/6/13; (e) rf05: 14/6/13.

***C3.2) Chapter 3.1.2. What is the period of simulation and spin up time? Why did you not use nested domains as in the US case?***

On August 13th all the air quality stations (i.e., Marche region air quality stations), reported the highest ozone daily eight-hour mean concentration value of the whole year. In light of this, we simulated a 6-day period with 2 days of spin up. The spin up is considered two days since we used both initial and boundary conditions for the meteorological and chemical parameters.

We initially did a nested domain over Italy with 4x4 km grid, but during the analysis of isoprene and ozone concentration, we chose to use the main domain to consider all values of Airbase across all Europe.

We added the following sentence in the Chapter 3.1.2.:

On August 13th, all the air quality stations (i.e., Marche region air quality stations), reported the highest ozone daily eight-hour mean concentration value of the whole year (**Error! Reference source not found.**-c). To represent the evolution of ozone peak event the simulations lasted 6 days, from August 10th (00:00 UTC) to August 16th (00:00 UTC), with 2 days of spin up for the model. A spin up time of 48 h is used for the chemistry to be consistent with the ambient conditions following past studies (Yerramilli et al., 2012; Zhang et al., 2009). The initial domain configuration used a nested domain over Italy with 4x4 km grid, but instead of focusing over the Marche region of Italy, we analyse the larger domain over Europe to explore the capabilities of the updated MEGAN algorithm for different vegetation types and chemistry regimes.

***C3.3) Line 280. How did you infer the conclusion regarding the mixing ratios inside the PBL versus the free troposphere from Figure 3?***

The x values represent the flight tracks and the relative aircraft-based measurements of isoprene concentrations under the Southern Oxidant and Aerosol Study (SOAS). When the isoprene concentrations decrease under 250 pptv (blue dots/lines in the plots), we have the aircraft in the free troposphere. In the contrary, when the values of isoprene mixing ratio are higher than 250 pptv, the color of dots are different from blue. See also Figure 15 panel a.

We added the reference to the Figure 15, rephrasing the sentence from:

From those flights, the first five (rf01-rf05) (It is not very easy to read. The unit of color bar is (mol km-2 h-1), and the isoprene emission factors are from the control simulation (i.e., with MEGAN v2.04). The x values represent the isoprene mixing ratios, (part per trillion by volume) along the flight tracks. We modified the caption from:

Figure 1), conducted mostly in the Southeast US to complement the SOAS objectives, were selected to estimate and evaluate the updates of the MEGAN code within the WRF-Chem model version 3.9. It is not very easy to read. The unit of color bar is (mol km-2 h-1), and the isoprene emission factors are from the control simulation (i.e., with MEGAN v2.04). The x values represent the isoprene mixing ratios, (part per trillion by volume) along the flight tracks. We modified the caption from:

Figure 1 shows that the aircraft sampled air in isoprene-rich emissions regions of United States, that the flights tracks show high isoprene mixing ratios when the aircraft was in the boundary layer, and therefore, the low isoprene mixing ratios occurred when the aircraft was in the free troposphere.

To:

For these flights, the aircraft sampled air in isoprene-rich emissions regions (Figure 3). Specifically, the flight tracks had high isoprene mixing ratios when the aircraft was in the boundary layer. The low isoprene mixing ratios occurred when the aircraft was above the boundary layer. For example, this trend can be observed for the time series of flight altitude (Figure 15 a) and measured isoprene concentration (Figure 15 c, black markers) for the second NOMADSS flight (rf02).

**C3.4) Line 291. How did you determine the 2 days is appropriate for the spin up time?**

We believe the 2 days spin up is appropriate to be consistent with the ambient conditions following the past studies which demonstrated that the WRF/Chem simulations are not very sensitive to the initial chemical conditions (Yerramilli et al., 2012; Zhang et al., 2009).

We do not add the information here since we explained already in the chapter 3.1.2, we added only reference.

**C4.1) Line 310. Why averaging the geopotential field over 6 days? How is this representing the evolution of the synoptic situation?**

We believe the 6-day average geopotential height map at 850 hPa could represent the evolution of the synoptic conditions since the presence of the intense geopotential height maximum (1520–1580 m), affecting the central part of the Mediterranean basin, is almost stationary for the duration of the period analyzed (i.e., 10-16 August). This behavior is also visible from the persistence of the high temperature across the southern Europe.

We changed the text, following your comment from:

We begin with evaluating the synoptic conditions predicted by the WRF-Chem simulations. The 6-day average geopotential height map at 850 hPa (Figure 5-a), shows the presence of an intense geopotential height maximum (1520–1580 m) affecting the central part of the Mediterranean basin.

***To:***

We begin with evaluating the synoptic conditions predicted by the WRF-Chem simulations. The 6-day average geopotential height map at 850 hPa (Figure 5-a), shows the presence of an intense geopotential height maximum (1520–1580 m), affecting the central part of the Mediterranean basin, in steady-state for the duration of the period analyzed.

***C4.2) Figure 5. I suggest plotting all the fields on the same map, i.e., plot the temperature as it is, but use contours with ticks for geopotential. Also, the surface pressure would be good to show on the same map also using contours.***

We modified the map using the contour line for geopotential height at 850 m over the temperature surface maps. We have chosen not to show the surface pressure.

[Figure]

**Figure 3: Comparison between the 6-day (August 10th – 15th, 2015) average geopotential height (m) at 850 hPa and mean temperature at 995 hPa, obtained with (a) NCAR/NCEP reanalysis and (b) the WRF-Chem model.**

***C4.4) Line 329. What does the expression "more staggered trend" mean?***

We changed the sentence from:

The map of PFTs percentage coverage reveals higher coverage of Needleleaf trees compared to Broadleaf, and Shrub and Bush in north-eastern Europe with values between 30–70 % and, with more staggered trend, in the north of Spain (i.e., the Cantabrian Mountains), Italy (i.e., Alps), Germany and in the most part of the Balkans peninsula (i.e., Carpathian Mountains)
To:

The map of PFTs percentage coverage reveals higher coverage of Needleleaf trees compared to Broadleaf, and Shrub and Bush in north-eastern Europe with values between 30–70 % and, with comparable trend, in the north of Spain (i.e., the Cantabrian Mountains), Italy (i.e., Alps), Germany and in the most part of the Balkans peninsula (i.e., Carpathian Mountains)

***C4.5) Line 337. In section 4.1 there is stated that BVOC observations are not available, so how do you analyze the "isoprene and alfa-pinene emissions"? Are the time series from the model?***

Yes, the time series of isoprene and alfa-pinene emissions (mol km$^{-2}$ hr$^{-1}$) are for different MEGAN algorithm configurations evaluated in 4 cities in Europe, without observational data included. In other words, it is only a comparison among the different MEGAN model changes.

***C4.6) Line 343-344. Why the MG and MGPFT gave the exact same emissions?***

We made 4 simulations considering:

1. **M2.04** -> the control run with no changes.

2. **MG** -> we updated all the gamma equations (LAI, PPFD, temperature, soil moisture and canopy environment), following G12 paper.

3. **MGPFT** -> here we updated the emission factor for 4 PFT, so we had two effects.

   a. Alpha-pinene emissions changed from previous simulation (i.e., MGPFT different from MG).

   b. Isoprene emissions did not change from previous simulation (i.e., MGPFT identical to MG). Later, we discovered that emission factor of Isoprene was considered directly from the pre-processor MEGAN; in

conclusion, the changes to PFT emission factor and PFT percentage, in the code, did not affect Isoprene.

4. **MGPFTISO** -> We forced the code to calculate the emission isoprene as the other compounds, instead of directly reading the value of emission factor from the database as the simulations before. This resulted that Isoprene emissions changed from previous simulation (i.e., MGPFTISO different from MGPFT), while other compounds remained the same (i.e., MGPFTISO identical to MGPFT).

We changed the explanation, following your comment, from:

2.1 Updates to MEGAN v2.04 in WRF-Chem

…..

In the present study, the changes made to the MEGAN algorithm implemented in WRF-Chem were the following: (i) update of the emission activity factors ($\gamma_i$), (ii) update of emission factor values for each plant functional type, and (iii) the assignment of the emission factor by PFT to isoprene.

To:

In the present study, we made four simulations with the following configurations. (i) The control run with no changes (M2.04). (ii) The updates to the emission activity factors (i.e., gamma equations for LAI, PPFD, temperature, soil moisture and canopy environment), following G12 paper (MG). (iii) The updates to the emission factor for 4 PFT (MGPFT). With this simulation we had two effects, firstly α-pinene emissions changed from the MG simulation to the MGPFT simulation, and secondly isoprene emissions did not change from the MG to the MGPFT simulation. In the MGPFT simulation, the changes to PFT emission factor and PFT percentage, in the code, did not affect isoprene as its emission factor was considered directly from the pre-processor MEGAN. (iv) We forced the code to calculate the isoprene emissions as the other compounds were determined, instead of directly reading the value of emission factor from the database as in the previous simulations. This resulted in isoprene emissions changing from previous simulation (i.e., MGPFTISO different from MGPFT), while α-pinene remained the same (i.e., MGPFTISO identical to MGPFT).

***C4.7) Lines 350-351. How did you determine the temperature range, is it the diurnal cycle? Is it from the model 995 hPa and if so, why choosing this level? What is the temperature used by MEGAN model, at which level and why?***

The temperature range was determinate by the analysis and comparison of WRF-Chem and the NCEP/NCAR reanalysis (Figure 5). We considered the temperature at 995 hPa since it is the first level included in the NCEP/NCAR reanalysis. The

temperature is considered as 6-day average, the duration of simulation.

The temperatures used by Megan model are:

a) T = T2 -> the instantaneous environment temperature at 2 m calculates from WRF-Chem.

b) T24 and T240 = "mtsa" -> climatological surface air temperature (K) (read in from file (wrfbiochemi_d<domain>)

**C4.8) Line 351. What this sentence means: "Kiev look like they may have experienced cloudiness based on the shape of the diurnal profile". Diurnal profile of which parameter? Measured or modeled? How it affects the emissions, is the cause the change in temperature or radiation?**

Sorry that this sentence was not clear. Based on the shape of the isoprene emissions in Figure 7, it appears that clouds (or some other meteorological factor) changed the smooth emissions diurnal profile to a more jagged shape. We have rewritten the sentence to the following.

On clear sky days, the isoprene emissions diurnal profile is smooth with a peak at midday. Clouds that form during the day can attenuate the solar radiation affecting the gamma-light parameter in the MEGAN calculation. In Figure 7, the more jagged diurnal profiles of isoprene emissions are likely due to cloudiness at different times of day.

**C4.9) Line 361. What is the meaning of the sentence "As with isoprene, the differences in the isoprene emission magnitudes are caused by the plant functional types, temperature and cloudiness for each city."?**

Thanks for noticing the error, it is a typo, we changed the sentence following your comment to:

As with isoprene, the differences in the α-pinene emission magnitudes are caused by the plant functional types, temperature, and cloudiness for each city.

**C4.10) Figure 11, add y and x labels on the figure.**

Thanks for the suggestion. We added the labels in the axis, and we tried to clarify the x and y axes in the figure caption, which has been rewritten to the following.

[Figure]

**Figure 13: Emission activity factors (y-axis, dimensionless) from M2.04 (M04) and M2.10 (M10) for different compound classes (1. Isoprene, 2. Myrcene, 3. Sabinene, 4. Limonene, 5. 3-Carene, 6. t-β-Ocimene, 7. β-Pinene, 8. α-Pinene, 9. Other Monoterpenes, 10. α-Farnesene, 11. β-Caryophyllene, 12. Other Sesquiterpenes, 13. 232-MBO, 14. Methanol, 15. Acetone, 16. Carbon Monoxide, 17. Nitric Oxide, 18. Bidirectional VOC, 19. Stress VOC and 20. other VOC). Each panel is for a different meteorological factor: (a) photosynthetic photon flux density ($\gamma_P$, GAMMA_P), (b) temperature ($\gamma_T$, GAMMA_T), (c) leaf age ($\gamma_{age}$, GAMMA_A), and (d) leaf area index ($\gamma_{LAI}$, GAMMA_LAI). The factors refer to the city of Genoa (Italy) on August 13th 885 (12:00 UTC), 2015.**

[Figure]

**Figure 14: Total emission activity factors (y-axis, dimensionless) from M2.04 (M04) and M2.10 (M10) for different compound classes (1. Isoprene, 2. Myrcene, 3. Sabinene, 4. Limonene, 5. 3-Carene, 6. t-β-Ocimene, 7. β-Pinene, 8. α-Pinene, 9. Other Monoterpenes, 10. α-Farnesene, 11. β-Caryophyllene, 12. Other Sesquiterpenes, 13. 232-MBO, 14. Methanol, 15. Acetone, 16. Carbon Monoxide, 17. Nitric Oxide, 18. Bidirectional VOC, 19. Stress VOC and 20. other VOC). Each panel is for different city: (a) Genoa (Italy), (b) Kiev (Ukraine), (c) Porto (Portugal), and (d) Zagreb (Croatia), on August 13th (12:00 UTC), 2015.**

**C4.11) Line 386, attitude?**

We would say that Zagreb and Genoa have the same behavior.

We replaced the sentence, follow your comment, from:

Genoa has seen its values double by updating the code from M2.04 to M2.10, particularly for γP, γT and γLAI (Figure 12), this attitude also applies to the city of Zagreb.

To:

Genoa has seen its values double by updating the code from M2.04 to M2.10, particularly for γP, γT and γLAI (Figure 12); Zagreb reports a similar trend.

**C4.12) Line 397. What do you mean by "Regardless of the location of the monitoring stations, the M2.10, MG, MGPFT runs show similar statistics for ozone….", How did you estimate the effect of the station location if you have one number for each station?**

Thanks for the suggestion, the sentence is not well expressed. We intended that the bias is the same independently of the station type (i.e., urban, suburban and rural).

We change the sentence, following your comment, from:

Regardless of the location of the monitoring stations, the M2.10, MG, MGPFT runs show similar statistics for ozone, with a consistent overestimation of ozone concentrations compared to the M2.04 run at each type of monitoring station.

To:

Regardless of the monitoring stations type (i.e., urban, suburban and rural), the M2.10, MG, MGPFT runs show similar statistics for ozone, with a consistent overestimation of its concentrations compared to the M2.04.

**C4.13) I see the general problem in the Chapter 4.1.3. regarding the evaluation of the model:**

   i) **the new version 2.10 and all the experiments show the worst results regarding the bias in O3,**

   ii) **the correlation coefficient is extremely low for the NO2 and especially for CO, but is it significant? What is the benefit of the new version if the results are worse?**

   iii) **How do you explain that there is a fairly good correlation between O3 measurements and modeled values, while CO and NO2 correlation is low?**

   i) We believe the higher O3 bias in M2.10 compared to M2.04 is that the BVOC emissions promotes the production of formaldehyde while NO2 concentrations are large (reaching 30 ppbv; Figure 13). The higher HCHO to NO2 ratios indicate higher O3 production (e.g., Souri et al., 2020). To give a better response, we will plot HCHO/NO2 in the final version, perhaps creating a scatter plot of O3 bias versus HCHO/NO2 for M2.04 and M2.10.

   ii) We do not think the low correlation of NO2, and CO is due to the changes in MEGAN modules, but rather to either the anthropogenic emission database or the biomass-burning emissions. As noted in Figure S4, CO concentrations decrease by 50-100 $\mu g \ m^{-3}$ over central Europe when

biomass burning emissions are not included, indicating a wide region affected by biomass-burning emissions. The underestimation of CO emissions from the fire emissions inventory likely causes the low CO correlation. This information is already included in the paper when discussing Figure S4.

iii) To fully understand the good correlation between O3 measurements and model results, the sources, and sinks of O3 need to be analyzed. Unfortunately, we did not adjust the biomass-burning, anthropogenic emissions, or global model initial and boundary conditions to improve the CO and NO2 model predictions. However, the aim of our paper is to note the impacts of using MEGAN v2.10 equations compared to MEGAN v2.04 keeping meteorology, anthropogenic emissions, and fire emissions the same. Our results suggest that further experimental and modeling studies should be conducted to address potential shortcomings in biogenic emission models.

**C4.14) What do you mean by "nitrogen dioxide coefficient correlation has no modifications"?**

We would state that between the different version of MEGAN we updated, the nitrogen dioxide correlation coefficient has not changed.

We updated the sentence, following you comment, from:

There are no remarkable modifications with the different MEGAN updates simulations: $O_3$ and CO have an increase of about 0.01–0.02 from the control run (M2.04) to the MEGAN updates simulations (MG, MGPFT, and M2.10); nitrogen dioxide coefficient correlation has no modifications.

To:

There are no remarkable modifications with the different MEGAN updates simulations: $O_3$ and CO have an increase of about 0.01–0.02 from the control run (M2.04) to the MEGAN updates simulations (MG, MGPFT, and M2.10), while nitrogen dioxide correlation coefficient has no variation between the different MEGAN updates.

**C4.15) Line 417-419, in the part of the sentence "; the highest biases in O3 tend to occur at locations where there are substantial discrepancies (up to 90 ugm-3)", the conclusion is inferred from where?**

Thanks for suggestion, we believe it is a typo. The sentence here does not make any sense, we simply removed it.

***C4.16) Lines 423-428, this part is confusing, how are the results in contrast with those of Jiang et al. (2019), I see no discussion upon low versus high mixing ratios overestimation/underestimation in your results?***

We revised the sentence according to your suggestion from:

However, the overestimation of the O3 concentrations compared to the Airbase data is about 20 µg m-3 (about 10 ppb) and up to about 40 µg m-3 (about 20 ppb) for the M2.04 and M2.10 simulations, respectively. The overestimation is visible for most of Europe, but it is more evident in central Europe (France, Germany, Switzerland, Austria and Northern Italy) and the south coast of the Iberian Peninsula. The results here contrast with those by Jiang et al. (2019), who found modelled ozone using the BVOC emission input from MEGAN v2.1 to be overestimated at low mixing ratios (20–50 ppb) and generally underestimated at mixing ratios above 50 ppb irrespective of the region of Europe considered.

To:

However, the overestimation of the O3 concentrations compared to the Airbase data is about 20 µg m-3 (about 10 ppb) and up to about 40 µg m-3 (about 20 ppb) for the M2.04 and M2.10 simulations, respectively. The overestimation is visible for most of Europe irrespective of the measured levels of O3 concentration, but it is more evident in central Europe (France, Germany, Switzerland, Austria and Northern Italy) and the south coast of the Iberian Peninsula. The results here contrast with those by Jiang et al. (2019), who found modelled ozone using the BVOC emission input from MEGAN v2.1 to be overestimated at low mixing ratios (20–50 ppb) and generally underestimated at mixing ratios above 50 ppb irrespective of the region of Europe considered.

***C4.17) Lines 428-429, what do you mean by "The NO2 (Figure 14-b) spatial resolution is not well represented in WRF-Chem, especially in north Europe…"***

The NO2 pattern represented in the Figure departs from the simulation results. In central part of Europe this result is well visible, therefore we deduced that the spatial resolution in not well represented by the model.

We update the sentence, following your comment, from:

The $NO_2$ (Figure 16-b) spatial resolution is not well represented in WRF-Chem, especially in north Europe (i.e., England, Belgium, Netherlands and North Germany), Northern Italy and Northeastern Spain.

To:

The NO$_2$ (Figure 14-b) spatial distribution is not well represented by WRF-Chem, especially in north Europe (i.e., England, Belgium, Netherlands and North Germany), Northern Italy and Northeastern Spain.

***C4.18) There are two main issues in chapter 4.2.1: i) there is no statistics calculated as r, RMSD, BIAS, ... ii) As in European case, the version v2.10 is shown to be worse than the older one, v2.4 (for isoprene and MACR).***

The reviewer is correct that the WRF-chem results with MEGAN 2.10 have a BVOC concentration bias increase over the Southeast US, showing poorer model evaluation than when using MEGAN 2.04. To better support these results, we will include a comprehensive statistical evaluation between WRF-Chem results and aircraft observations in the final version of the paper.

***C4.19) Lines 461-463, how are the M2.04 isoprene values overpredicted by factor 5 (Fig. 5)?***

We considered when the aircraft goes in the PBL. The difference between M2.04 and M2.10 isoprene value reach difference values up to 10 ppb.

We update the sentence, following your comment, from:

Comparison of M2.04 and M2.10 simulations to aircraft observations shows that isoprene (Figure 15-c) mixing ratios agree well with measured isoprene for the M2.04 simulation but are overpredicted by up to a factor of 5 in the PBL.

To:

Comparison of M2.04 and M2.10 simulations to aircraft observations shows that isoprene (Figure 15-c) mixing ratios agree well with measured isoprene for the M2.04 simulation but are overpredicted by up to 10 ppbv in the PBL.

***C4.20) Lines after 482. The discussion is not very clear, what is the conclusion in the end, is the problem in the Megan or the PBL schemes? If the PBL scheme is consistent with meteorological observations, and you use the same meteo setup in all experiments, isn't there a problem in v2.10?***

***C4.21) Lines 488-490. What is the meaning of the sentence "However, differences between modelled and aircraft data likely do not depend on boundary layer meteorological variables as measurements flights generally take place under***

*weather conditions and boundary layer heights scarcely affected by boundary layer mixing phenomena (Travis et al., 2016)."?*

Response to C4.20) and C4.21)

The problem of the increase of bias for the concentration values of v2.10 is not due to the PBL scheme, but to the change we made in the code.

We update the sentence, following your comment, from:

Nevertheless, for our study this is likely not the cause, and the ozone overprediction is mainly due to the isoprene emission changes. According to large-eddy simulations (Kim et al., 2016; Li et al., 2016; Ouwersloot et al., 2011) and measurement-model analysis (Kaser et al., 2015) the effects of physical separation of isoprene and OH in the PBL depends on chemistry-turbulence interactions and scale dependent heterogeneity of isoprene emissions, with potential implications on CTMs. The differences observed between measured and modelled isoprene mixing ratios along flight tracks may depend on the complex interaction between chemical reactions involving isoprene and turbulence within the PBL (Zhao et al., 2016). However, differences between modelled and aircraft data likely do not depend on boundary layer meteorological variables as measurements flights generally take place under weather conditions and boundary layer heights scarcely affected by boundary layer mixing phenomena (Travis et al., 2016).

To:

Nevertheless, for our study this is likely not the cause, and the ozone overprediction is mainly due to the isoprene emission changes. According to large-eddy simulations (Kim et al., 2016; Li et al., 2016; Ouwersloot et al., 2011) and measurement-model analysis (Kaser et al., 2015) the effects of physical separation of isoprene and OH in the PBL depends on chemistry-turbulence interactions and scale dependent heterogeneity of isoprene emissions, with potential implications on CTMs. The differences observed between measured and modelled isoprene mixing ratios along flight tracks may depend on the complex interaction between chemical reactions involving isoprene and turbulence within the PBL (Zhao et al., 2016). However, measurements flights generally take place under weather conditions and boundary layer heights scarcely affected by boundary layer mixing phenomena (Travis et al., 2016).Therefore, differences between modelled and aircraft data, that were observed in the present study, likely do not depend on simulated values of boundary layer meteorological variables.

***C5.1) Lines 500-505. The conclusion is missing, why is v2.10 having higher bias?***

The version 2.10 has higher bias as we change the MEGAN emission algorithms following Guenther et al. (2012), for the emission activity factor and emission factor for PFT.

We updated the last part of the conclusion, following your comments, from:

Comparisons between modelled and surface measured (Airbase database) ozone concentrations showed values of the correlation coefficients in the range from 0.78 to 0.86, with higher values for the rural monitoring stations compared to the urban and suburban ones as well as for the M2.10 run compared to the M2.04 simulation. Moreover, the spatial distribution of modelled $O_3$ concentrations represented well the observed values, regardless of the simulations considered (M2.04, MG, MGPFT, and M2.10). However, magnitude differences were observed in both M2.04 and M2.10 simulations, with an overestimation of the $O_3$ concentrations compared to the Airbase data by about 20 μg m$^{-3}$ (10 ppb) and up to about 40 μg m$^{-3}$ (20 ppb), respectively.

To:

Comparisons between modelled and surface measured (Airbase database) ozone concentrations showed values of the correlation coefficients in the range from 0.78 to 0.86, with higher values for the rural monitoring stations compared to the urban and suburban ones. Higher correlation coefficients were also higher in the M2.10 run compared to the M2.04 simulation. Moreover, the spatial distribution of modelled $O_3$ concentrations represented well the observed values, regardless of the simulations considered (M2.04, MG, MGPFT, and M2.10). However, magnitude differences were observed in both M2.04 and M2.10 simulations, with an overestimation of the $O_3$ concentrations compared to the Airbase data by about 20 μg m$^{-3}$ (10 ppb) and up to about 40 μg m$^{-3}$ (20 ppb), respectively. The higher O3 bias in M2.10 compared to M2.04 is believed to be due to increased formaldehyde concentrations which is a product of the BVOC chemistry.

***C5.2) Lines 505-510. Statistical evaluation is not performed, only comparison of the diagrams***

The statistical evaluation will be performed in the final version.

***C5.3) Lines 519-530. What do you mean by stating the new model has more flexibility? The idea of verification the results with satellite observation is valuable, but will this improve the model? Can you be more specific on what should be improved in the model based on the results from this paper.***

We rephrase the sentence, following your comment, from:

[revised manuscript text omitted]